# Adaptive Batch Sizes Using Non-Euclidean Gradient Noise Scales for Stochastic Sign and Spectral Descent

**Hiroki Naganuma** [1] [2]  **Shagun Gupta** [3]  **Youssef Briki** [2]  **Ioannis Mitliagkas** [1] [2]  **Irina Rish** [1] [2]
**Parameswaran Raman** [3]  **Hao-Jun Michael Shi** [3]

## Abstract

To maximize hardware utilization, modern machine learning systems typically employ large constant or manually tuned batch size schedules, relying on heuristics that are brittle and costly to tune. Existing adaptive strategies based on gradient noise scale (GNS) offer a principled alternative. However, their assumption of SGD's Euclidean geometry creates a fundamental mismatch with popular optimizers based on generalized norms, such as signSGD / Signum ($\ell_\infty$) and stochastic spectral descent (specSGD) / Muon ($\mathcal{S}_\infty$). In this work, we derive gradient noise scales for signSGD and specSGD that naturally emerge from the geometry of their respective dual norms. To practically estimate these non-Euclidean metrics, we propose an efficient variance estimation procedure that leverages the local mini-batch gradients on different ranks in distributed data-parallel systems. Our experiments demonstrate that adaptive batch size strategies using non-Euclidean GNS enable us to match the validation loss of constant-batch baselines while reducing training steps by up to 66% for Signum and Muon on a 160 million parameter Llama model.

## 1. Introduction

As model architectures and datasets scale exponentially, maximizing hardware utilization has become a critical objective in training modern machine learning models. This is typically achieved by scaling the batch size, which improves GPU throughput by processing a greater number of samples in parallel and reduces the total number of optimizer steps, ultimately shortening the end-to-end wall-clock training time (Smith et al., 2018; McCandlish et al., 2018). Despite these benefits, constant large batch sizes suffer from diminishing returns in per-sample improvement. This reduction in sample efficiency compromises model quality under fixed training budgets (Keskar et al., 2017; Smith et al., 2018; Shallue et al., 2019; Nado et al., 2021; Marek et al., 2026).

To manage this trade-off, practitioners employ heuristic schedules that increase the batch size during training, such as through *stage-wise* (Rae et al., 2021; Hoffmann et al., 2022; Chowdhery et al., 2023; Grattafiori et al., 2024; Groeneveld et al., 2024) or *linear ramp-ups* (Brown et al., 2020; Smith et al., 2022; Liu et al., 2024). These methods attempt to balance small-batch efficiency with large-batch throughput; yet, they are brittle and demand significant tuning. Without a principled, computable metric to guide the schedule, practitioners lack a reliable way of predicting when increasing the batch size will compromise sample efficiency.

A more principled alternative involves dynamically choosing the batch size by monitoring the *gradient noise scale* (GNS), a metric that quantifies the relative error in the gradient estimate (Byrd et al., 2012; McCandlish et al., 2018; Chen et al., 2025). By adjusting the batch size to track this signal, one can avoid the diminishing return associated with exceeding the *critical batch size* (CBS). Although various formalizations of the CBS exist, they share the objective of identifying the threshold above which further increases in batch size yield negligible gains in per-sample improvement. Theoretically, this threshold is defined as a constant multiple of the GNS (see Section 3). In this work, we focus on designing GNS metrics to characterize and implement adaptive batch size strategies.

While previous research derived the CBS and GNS based on the descent lemma for the stochastic gradient method (SGD) (McCandlish et al., 2018) and its underlying Euclidean geometry, modern vision and language workloads favor optimizers that operate in *non-Euclidean geometries*, such as sign-based methods (Bernstein et al., 2018) and spectral methods (Carlson et al., 2015a;b; Bernstein & Newhouse, 2024b; Jordan et al., 2024). Furthermore, popular

[1]Mila, Montreal, Canada [2]Université de Montréal, Montreal, Canada [3]Meta Platforms, Menlo Park, California, USA. Correspondence to: Hiroki Naganuma <naganuma.hiroki@mila.quebec>, Shagun Gupta <shagun@meta.com>.

*Proceedings of the 43rd International Conference on Machine Learning*, Seoul, South Korea. PMLR 306, 2026. Copyright 2026 by the author(s).

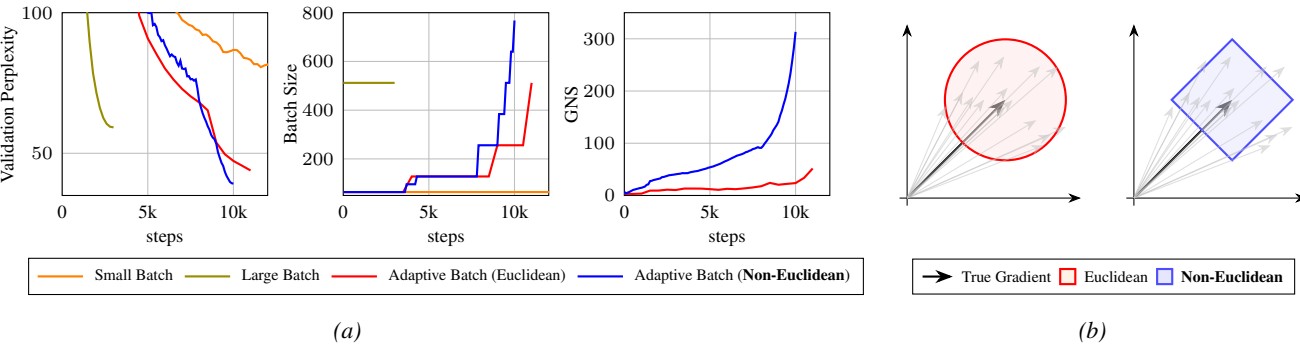

*(a)*        *(b)*

*Figure 1.* In **1a (left)**, we plot the validation perplexity, batch size, and the exponential moving average of the GNS over steps when training the 160M parameter Llama 3 model for 3.2B tokens on C4 data using signSGD. We compare small-batch ($B = 64$), large-batch ($B = 512$), and adaptive batch size strategies based on Euclidean and non-Euclidean GNS. The plots highlight the improved efficiency of using non-Euclidean GNS measurements for signSGD. In **1b (right)**, we depict how enforcing different GNS ($\ell_2$ versus $\ell_1$) changes the geometry of the allowable region for stochastic gradients (with high probability).

optimizers like Adam (Kingma, 2014), Shampoo (Gupta et al., 2018), and SOAP (Vyas et al., 2025; Eschenhagen et al., 2026) have been increasingly interpreted from this non-Euclidean lens, sharing fundamental geometric properties with sign and spectral updates (Balles & Hennig, 2018; Orvieto & Gower, 2026). Consequently, the current literature suffers from a fundamental misalignment: using Euclidean-based GNS metrics to determine batch sizes dynamically for non-Euclidean optimizers.

To bridge this gap, we derive metrics for stochastic sign descent (signSGD) and stochastic spectral descent (specSGD) that measure the gradient noise scale in the dual norm of the optimizer's geometry (i.e., $\ell_1$ norm for sign-based optimizers and nuclear norm for spectral optimizers). When analyzed with respect to their corresponding geometry, the primary source of error in signSGD and specSGD is induced by the *bias* in the search direction, which can be bounded by the expected dual norm of the gradient error. Leveraging these refined GNS statistics yields substantial empirical benefits in practice, as illustrated in Figure 1.

Efficiently estimating GNS metrics for adaptive batch sizes remains a challenge due to high memory and computational overhead. Prior approaches that calculate variance using individual samples are memory-intensive and untenable for large-scale training (Bollapragada et al., 2018a; Chen et al., 2025). Alternative variance estimators that compare local and global batches are practical but high variance since they rely on a single sample (McCandlish et al., 2018; Merrill et al., 2026). Instead, we propose to use the local mini-batch gradients across data-parallel ranks as multiple independent samples, appending partial statistics to the gradient `AllReduce` operation required during distributed training. This approach leverages the distributed frameworks (e.g., DDP (Li et al., 2020) and FSDP (Zhao et al., 2023)) already intrinsic to most large machine learning systems to provide a reliable real-time GNS signal.

**Contributions.** In this work, we derive *non-Euclidean GNS* metrics for sign-based (signSGD, Signum) and spectral optimizers (specSGD, Muon). We propose an adaptive batch size strategy and a scalable variance estimation procedure tailored for distributed training setups. We demonstrate that adaptive batch size strategies guided by these non-Euclidean metrics yield up to a 66% *reduction in optimizer steps* for language and vision tasks. Crucially, our schedules match the validation loss of constant-batch baselines while significantly improving training efficiency.

**Conflict of Interest Disclosure.** The authors declare that they have no relevant financial or substantive conflicts of interest to disclose.

## 2. Background

We formulate the training objective as the expected risk minimization problem:

$$\min_{x \in \mathbb{R}^d} \mathcal{L}(x) = \mathbb{E}_{\xi \sim \mathcal{D}} \left[ \ell(x; \xi) \right] \tag{1}$$

where $\xi$ is drawn from an underlying stationary distribution $\mathcal{D}$ with support $\Xi$, $\ell : \mathbb{R}^d \times \Xi \to \mathbb{R}$ is the loss function evaluated on sample $\xi$, and $\mathbb{E}$ denotes the expectation with respect to $\mathcal{D}$.

### 2.1. Stochastic Generalized Steepest Descent

We consider stochastic mini-batch variants of the *generalized steepest descent method*. At each iteration $k$, the method samples a mini-batch $\mathcal{S}_k$ of $B_k$ i.i.d. samples from $\mathcal{D}$, then computes the mini-batch gradient $g_k = \frac{1}{B_k} \sum_{\xi \in \mathcal{S}_k} \nabla \ell(x_k; \xi) \in \mathbb{R}^d$. The current iterate $x_k$ is up-

dated via:

$$x_{k+1} = x_k - \eta_k p_k, \quad \text{where} \quad p_k = \arg\max_{\|p\|\leq 1} \langle g_k, p \rangle. \quad (2)$$

The search direction $p_k$ is the steepest descent direction for the stochastic gradient estimate over the chosen norm $\|\cdot\|$ (with corresponding dual norm $\|\cdot\|_*$). As noted in Wright et al. (1999); Boyd & Vandenberghe (2004) and Beck (2017), different choices for primal and dual norms ($\|\cdot\| / \|\cdot\|_*$) recover standard optimization algorithms:

- $\ell_2$ norm ($\|\cdot\|_2 / \|\cdot\|_2$): Yields normalized SGD, where $p_k = g_k/\|g_k\|_2$.

- $\ell_\infty$ norm ($\|\cdot\|_\infty / \|\cdot\|_1$): Yields stochastic sign descent (signSGD), where $p_k = \text{sign}(g_k)$.

- Spectral or Schatten-$\infty$ norm ($\|\cdot\|_{\mathcal{S}_\infty} / \|\cdot\|_{\mathcal{S}_1}$): For matrix variables $X \in \mathbb{R}^{m \times n}$, yields stochastic spectral descent (specSGD) with $P_k = \text{matsign}(G_k) = U_k V_k^\top$.[1]

Here, the matrix sign function is defined via the reduced SVD $G_k = U_k \Sigma_k V_k^\top$ and taking the sign function of the singular values, i.e., $P_k = U_k V_k^\top$. Geometrically, $P_k$ corresponds to the unitary factor of the polar decomposition of $G_k$, which is the closest semi-orthogonal matrix to $G_k$ in Frobenius norm.[2]

Other popular optimizers such as Adam (Kingma, 2014) and signSGD with momentum (also known as Signum) (Bernstein et al., 2018) as well as Shampoo (Gupta et al., 2018) and Muon (Jordan et al., 2024) are known to reduce to signSGD and specSGD as special cases, respectively (Balles & Hennig, 2018; Bernstein & Newhouse, 2024b; Orvieto & Gower, 2026). While specSGD was originally suggested for training neural networks by Carlson et al. (2015a;b), it has recently resurged in popularity due to its connection to the theory of modular duality (Bernstein & Newhouse, 2024a) and the Muon optimizer (Jordan et al., 2024; Liu et al., 2025), which utilizes momentum and the Newton-Schulz iteration to efficiently approximate the semi-orthogonalization of the gradient. These interpretations motivate our study of adaptive batch sizes for these fundamental methods.

## 2.2. Critical Batch Size and Gradient Noise Scale

Training with large batch sizes is known to degrade model quality and sample efficiency after exceeding a certain point,

often referred to as the *critical batch size* (CBS) (Keskar et al., 2017; Shallue et al., 2019; McCandlish et al., 2018). Theoretically, the CBS is defined as the batch size that minimizes the total computational complexity (e.g., stochastic first-order oracle complexity). Recent work has applied this framework to derive closed-form CBS expressions for specific optimizers like Muon (Sato et al., 2025).

To mitigate this degradation in practice, various heuristics have been proposed. Most notable among these are learning rate warmup and scaling rules: *linear scaling* (Goyal et al., 2017; Smith et al., 2018) and *square root scaling* (Krizhevsky, 2014; Hoffer et al., 2017). Additionally, layerwise adaptive methods like LARS (You et al., 2017) and LAMB (You et al., 2020) were developed to stabilize training by adjusting per-layer learning rates based on the ratio of weight to gradient norms.

Despite the success of these techniques, training with a fixed batch size inevitably encounters a regime of diminishing returns. Since the number of samples consumed by each optimizer step scales with the batch size, the optimizer must make more progress per-step to maintain sample efficiency. While this is typically achieved by scaling the learning rate to leverage the gradient's reduced variance, this strategy is limited by the deterministic properties of the loss landscape, i.e., the maximum curvature (Wright et al., 1999; Mandt et al., 2017; Gilmer et al., 2022). Consequently, further reducing the gradient variance by increasing the batch size beyond the CBS provides little additional benefit per step.

**Euclidean gradient noise scale.** The seminal work by McCandlish et al. (2018) derives a formal notion of CBS based on the optimal expected single-step improvement in the loss function for (un-normalized) SGD.[3] Let $C_k \in \mathbb{S}_+^d$ denote the gradient covariance matrix at step $k$:

$$C_k = \mathbb{E}_k[(\nabla \ell(x_k; \xi) - \nabla \mathcal{L}(x_k))(\nabla \ell(x_k; \xi) - \nabla \mathcal{L}(x_k))^\top]$$

where $\mathbb{E}_k[\cdot] = \mathbb{E}[\cdot \mid x_k]$ denotes the conditional expectation given $x_k$. By leveraging a quadratic approximation of the loss function (assuming $\nabla^2 \mathcal{L}(x_k) \approx I$ (McCandlish et al., 2018)):

$$\mathcal{L}(x_{k+1}) = \mathcal{L}(x_k) - \eta_k \langle \nabla \mathcal{L}(x_k), g_k \rangle + \frac{\eta_k^2}{2} \|g_k\|_2^2, \quad (3)$$

one can derive the *expected one-step improvement* for SGD:

$$\Delta\mathcal{L}(x_k, \eta_k, \|\cdot\|_2, B_k) = \mathbb{E}_k[\mathcal{L}(x_k) - \mathcal{L}(x_{k+1})]$$
$$= \eta_k \|\nabla \mathcal{L}(x_k)\|_2^2 - \frac{\eta_k^2}{2}\left(\|\nabla \mathcal{L}(x_k)\|_2^2 + \frac{\text{tr}(C_k)}{B_k}\right). \quad (4)$$

---

[1]The Schatten-1 norm is the nuclear norm $\|G_k\|_{\mathcal{S}_1} = \text{tr}(C_k)$.

[2]One may also consider un-normalized variants of these methods, which additionally multiply $p_k$ by the dual norm of the gradient $\|g_k\|_*$. In the deterministic setting, these methods share a unified analysis based on smoothness with respect to arbitrary norms $\|\cdot\|$ (Balles et al., 2020). However, besides SGD, these variants are not commonly used in practice.

[3]Note that CBS is an overloaded term: In the systems literature, it often refers to the maximum efficient batch size for an entire training run (Shallue et al., 2019). Here, we refer to the *instantaneous* CBS at a specific optimization step $k$.

Here, the equality follows from the bias-variance decomposition $\mathbb{E}_k[\|g_k\|_2^2] = \|\nabla\mathcal{L}(x_k)\|_2^2 + \mathbb{E}_k[\|g_k - \nabla\mathcal{L}(x_k)\|_2^2]$.

Maximizing the expected improvement in Equation (4) with respect to $\eta_k$ yields the optimal learning rate:

$$\eta_k^*(B_k) = \left(1 + \frac{1}{B_k} \cdot \frac{\text{tr}(C_k)}{\|\nabla\mathcal{L}(x_k)\|_2^2}\right)^{-1}. \quad (5)$$

Plugging $\eta_k^*(B_k)$ into Equation (4) yields the optimal expected improvement, which is proportional to $\eta_k^*(B_k)$. Intrinsic to both quantities is the *Euclidean GNS*, denoted as $\mathcal{B}_{\ell_2}(x_k)$, which governs the trade-off between the batch size and the realizable progress:

---
**Euclidean ($\ell_2$) GNS for SGD**

$$\mathcal{B}_{\ell_2}(x_k) = \frac{\text{tr}(C_k)}{\|\nabla\mathcal{L}(x_k)\|_2^2}. \quad (6)$$

---

Practitioners utilize this metric ($\mathcal{B} = \mathcal{B}_{\ell_2}$) to select the batch size:

---
**GNS → Batch Size Rule**

$$B_k = \theta^{-2}\mathcal{B}(x_k), \quad \theta \in (0, 1). \quad (7)$$

---

The tolerance parameter $\theta \in (0, 1)$ enforces a constant noise-to-signal ratio in the gradient estimate. This rule is equivalent to the *norm test* proposed in the classical optimization literature to ensure linear convergence for strongly convex functions (Byrd et al., 2012):

$$\mathbb{E}[\|g_k - \nabla\mathcal{L}(x_k)\|_2^2] \leq \theta^2\|\nabla\mathcal{L}(x_k)\|_2^2.$$

Alternative adaptive strategies utilizing gradient diversity (Chen et al., 2025), angular distances (Bollapragada et al., 2018a;b; Bahamou & Goldfarb, 2019; Xu et al., 2020), local branch training (Merrill et al., 2026), composite metrics (Belias et al., 2025), and the learning rate schedule (Devarakonda et al., 2017; Smith et al., 2018) have also been proposed.

While this analysis provides a robust framework for SGD relying on the Euclidean structure in (4), recent attempts to apply it to non-Euclidean methods (Gray et al., 2024; Lau et al., 2024a) inherently ignore the optimizer's geometry. In the following section, we show how to formally extend these ideas to signSGD and specSGD.

## 3. Non-Euclidean GNS

To generalize the theory of CBS and GNS beyond Euclidean settings, we observe that the *dual norm* of the optimizer is the natural metric for quantifying noise in generalized stochastic steepest descent. This relationship is obscured in the Euclidean case because the $\ell_2$ norm is its own dual.

For signSGD and specSGD, the stochastic steepest descent direction is *biased*, i.e., $\mathbb{E}_k[p_k] \neq \arg\max_{\|p\|\leq 1}\langle\nabla\mathcal{L}(x_k), p\rangle$. We control this bias term by controlling the GNS. To make this problem tractable, we derive the GNS by maximizing a lower bound on the expected single-step improvement:

**Lemma 3.1.** *Let $\|\cdot\|$ be a general norm and $\|\cdot\|_*$ be its associated dual norm. The stochastic steepest descent direction $p_k = \arg\max_{\|p\|\leq 1}\langle g_k, p\rangle$ satisfies:*

$$\mathbb{E}_k\left[\langle\nabla\mathcal{L}(x_k), p_k\rangle\right] \geq \|\nabla\mathcal{L}(x_k)\|_* - \mathbb{E}_k\left[\|\nabla\mathcal{L}(x_k) - g_k\|_*\right].$$

The proof for Lemma 3.1 is provided in Appendix A.

Again using a quadratic approximation of the loss function, we generalize Equation (3) as:

$$\mathcal{L}(x_{k+1}) = \mathcal{L}(x_k) - \eta_k\langle\nabla\mathcal{L}(x_k), p_k\rangle + \frac{\eta_k^2}{2}\langle p_k, p_k\rangle. \quad (8)$$

Taking the expectation of Equation (8) given $x_k$ and using Lemma 3.1, we obtain the key inequality:

$$\begin{aligned}
&\Delta\mathcal{L}(x_k, \eta_k, \|\cdot\|, B_k) \\
&\geq \eta_k\|\nabla\mathcal{L}(x_k)\|_* - \eta_k\mathbb{E}_k\left[\|\nabla\mathcal{L}(x_k) - g_k\|_*\right] \\
&\quad - \frac{\eta_k^2}{2}\mathbb{E}_k\left[\langle p_k, p_k\rangle\right].
\end{aligned} \quad (9)$$

Since the update has unit norm, the quadratic term $\mathbb{E}_k\left[\langle p_k, p_k\rangle\right]$ is constant and easily evaluated (e.g., $d$ for signSGD). Therefore, the primary challenge is in evaluating the expectation of the gradient estimation error measured in the dual norm, $\mathbb{E}_k\left[\|\nabla\mathcal{L}(x_k) - g_k\|_*\right]$. We characterize this error for signSGD and specSGD to derive their respective GNS.

### 3.1. Stochastic Sign Descent (signSGD)

As seen in Section 2.1, the search direction $p_k = \text{sign}(g_k)$ corresponds to the steepest descent direction under the $\ell_\infty$ norm. Let $\sigma_k \in \mathbb{R}^d$ be the vector consisting of the component-wise standard deviations of the single-sample gradient $\nabla\ell(x_k; \xi)$. That is, for each component $i = 1, ..., d$:

$$[\sigma_k]_i^2 = \mathbb{E}\left[([\nabla\ell(x_k; \xi) - \nabla\mathcal{L}(x_k)]_i)^2\right].$$

To bound the key inequality in Equation (9), we utilize the following inequality which bounds the expected $\ell_1$ norm of the gradient error (Bernstein et al., 2018):[4]

$$\mathbb{E}_k\left[\|\nabla\mathcal{L}(x_k) - g_k\|_1\right] \leq \frac{\|\sigma_k\|_1}{\sqrt{B_k}}. \quad (10)$$

---

[4]We provide complete statements and proofs in Appendix A.

The dependence on $\sqrt{B_k}$ ensures that the error vanishes as the batch size increases. One could obtain a tighter bound on the error by estimating $\mathbb{E}_k\left[\|\nabla\mathcal{L}(x_k) - g_k\|_1\right]$ directly.

Substituting Equation (10) and the identity $\langle\text{sign}(g_k), \text{sign}(g_k)\rangle \leq d$ into Equation (9), we obtain the specific lower bound for signSGD:

$$\begin{aligned}
&\Delta\mathcal{L}(x_k, \eta_k, \|\cdot\|_\infty, B_k) \\
&\quad \geq \eta_k\left(\|\nabla\mathcal{L}(x_k)\|_1 - \frac{\|\sigma_k\|_1}{\sqrt{B_k}}\right) - \frac{d\eta_k^2}{2}.
\end{aligned} \quad (11)$$

The learning rate that maximizes this lower bound is:

$$\eta_k^*(B_k) = \frac{\|\nabla\mathcal{L}(x_k)\|_1}{d}\left(1 - \frac{1}{\sqrt{B_k}}\cdot\frac{\|\sigma_k\|_1}{\|\nabla\mathcal{L}(x_k)\|_1}\right). \quad (12)$$

The structure in $\eta_k^*(B_k)$ reveals the *Manhattan ($\ell_1$) GNS*:

> **Manhattan ($\ell_1$) GNS for signSGD**
>
> $$\mathcal{B}_{\ell_1}(x_k) = \frac{\|\sigma_k\|_1^2}{\|\nabla\mathcal{L}(x_k)\|_1^2}. \quad (13)$$

The $\ell_1$ GNS fundamentally differs from the Euclidean $\ell_2$ GNS in measuring the gradient norm via the $\ell_1$ norm and noise via the squared sum of standard deviations $\|\sigma_k\|_1^2$. Geometrically, $\ell_1$ GNS controls the probability of correct sign assignment for coordinate-wise updates.

While standard signSGD is known to converge only to a neighborhood (determined by $\|\sigma_k\|_1$) when using a fixed batch size and learning rate (Karimireddy et al., 2019), adhering to this adaptive schedule ensures the noise term in Equation (11) remains controlled. This effectively behaves as a variance reduction mechanism similar to error feedback or momentum in Signum (Bernstein et al., 2018) (see Appendix A). Substituting $B_k = \theta^{-2}\mathcal{B}_{\ell_1}(x_k)$ into (11) yields the factor $(1 - \theta)^2$, which appears explicitly in our convergence results (see Theorem A.4).

## 3.2. Stochastic Spectral Descent (specSGD)

Similar to signSGD, specSGD for matrix weights $X_k \in \mathbb{R}^{m\times n}$ (assuming $m \leq n$) defines the search direction $P_k = \text{matsign}(G_k)$, which corresponds to the steepest descent direction under the Schatten-$\infty$ (spectral) norm. The matrix sign is given by $P_k = U_k V_k^\top$, where $G_k = U_k\Sigma_k V_k^\top$ is the reduced SVD of the gradient. Let $C_{\text{row},k} \in \mathbb{S}_+^m$ be the

row-wise aggregated covariance matrix[5], i.e.,

$$C_{\text{row},k} = \mathbb{E}_k\left[\begin{array}{c}(\nabla\ell(X_k;\xi) - \nabla\mathcal{L}(X_k)) \\ \cdot\left(\nabla\ell(X_k;\xi) - \nabla\mathcal{L}(X_k)\right)^\top\end{array}\right].$$

The analysis is strictly analogous to the signSGD setting, replacing vector $\ell_1$ norms with nuclear (Schatten-1) norms. Specifically, we use a similar error bound:

$$\mathbb{E}_k\left[\|\nabla\mathcal{L}(X_k) - G_k\|_{\mathcal{S}_1}\right] \leq \frac{\|C_{\text{row},k}^{1/2}\|_{\mathcal{S}_1}}{\sqrt{B_k}}. \quad (14)$$

Here, the nuclear norm $\|C_{\text{row},k}^{1/2}\|_{\mathcal{S}_1}$ represents the sum of the singular values of the row-wise covariance matrix.

Substituting Equation (14) into Equation (9), and noting that the inner product of the search direction is $\langle P_k, P_k\rangle = \text{tr}(V_k U_k^\top U_k V_k^\top) = \text{tr}(I_r) = r \leq m$, where $\text{rank}(G_k) = r$, we obtain:

$$\begin{aligned}
&\Delta\mathcal{L}(X_k, \eta_k, \|\cdot\|_{\mathcal{S}_\infty}, B_k) \\
&\quad \geq \eta_k\left(\|\nabla\mathcal{L}(X_k)\|_{\mathcal{S}_1} - \frac{\|C_{\text{row},k}^{1/2}\|_{\mathcal{S}_1}}{\sqrt{B_k}}\right) - \frac{r\eta_k^2}{2}.
\end{aligned} \quad (15)$$

Maximizing this bound yields the optimal learning rate and expected improvement:

$$\eta_k^*(B_k) = \frac{\|\nabla\mathcal{L}(x_k)\|_{\mathcal{S}_1}}{r}\left(1 - \frac{1}{\sqrt{B_k}}\cdot\frac{\|C_{\text{row},k}^{1/2}\|_{\mathcal{S}_1}}{\|\nabla\mathcal{L}(x_k)\|_{\mathcal{S}_1}}\right). \quad (16)$$

This admits the Nuclear ($\mathcal{S}_1$) GNS, which mimics the $\ell_1$ GNS in Equation (13):

> **Nuclear ($\mathcal{S}_1$) GNS for specSGD**
>
> $$\mathcal{B}_{\mathcal{S}_1}(x_k) = \frac{\|C_{\text{row},k}^{1/2}\|_{\mathcal{S}_1}^2}{\|\nabla\mathcal{L}(X_k)\|_{\mathcal{S}_1}^2}. \quad (17)$$

**Analogous properties.** Most of the geometric intuition and convergence guarantees derived for signSGD transfer to specSGD, substituting $\ell_\infty/\ell_1$ norms with their spectral/nuclear counterparts. When $B_k \ll \mathcal{B}_{\mathcal{S}_1}$, the noise dominates the gradient's spectral structure, causing the unitary factor $U_k V_k^\top$ to point in a random direction effectively uncorrelated with the true spectral geometry. Ensuring $B_k \gg \mathcal{B}_{\mathcal{S}_1}$

---

[5]The column-wise aggregated covariance matrix $C_{\text{col},k} = \mathbb{E}_k[(\nabla\ell(X_k;\xi) - \nabla\mathcal{L}(X_k))^\top(\nabla\ell(X_k;\xi) - \nabla\mathcal{L}(X_k))]$ yields an equivalent nuclear norm $\|C_{\text{row},k}^{1/2}\|_{\mathcal{S}_1} = \text{tr}(C_{\text{row},k}^{1/2}) = \text{tr}(C_{\text{col},k}^{1/2}) = \|C_{\text{col},k}^{1/2}\|_{\mathcal{S}_1}$. Therefore, if $m > n$, we can use $C_{\text{col},k} \in \mathbb{S}_+^n$ instead to save memory.

*Table 1.* Gradient Noise Scale (GNS) for SGD, signSGD, and specSGD expressed as a squared ratio of dual norms. Note the consistent pattern: primal norm $\| \cdot \|$, dual norm $\| \cdot \|_*$, GNS $\propto (\|\text{Noise}\|_* / \|\text{Signal}\|_*)^2$.

| Method | Geometry | GNS |
|---|---|---|
| SGD | $\ell_2$ | $\dfrac{\text{tr}(C_k)}{\|\nabla\mathcal{L}(x_k)\|_2^2}$ |
| signSGD | $\ell_\infty$ | $\dfrac{\|\sigma_k\|_1^2}{\|\nabla\mathcal{L}(x_k)\|_1^2}$ |
| specSGD | $\mathcal{S}_\infty$ | $\dfrac{\|C_{\text{row},k}^{1/2}\|_{\mathcal{S}_1}^2}{\|\nabla\mathcal{L}(X_k)\|_{\mathcal{S}_1}^2}$ |

aligns the search direction with the true gradient's dominant singular subspaces. Consequently, the same adaptive schedule introduces the identical $(1 - \theta)^2$ factor into our convergence results (see Theorem A.5).

Although we use normalized steepest descent for signSGD and specSGD, the dual norm scaling ($\|\nabla\mathcal{L}(x_k)\|_1$ and $\|\nabla\mathcal{L}(X_k)\|_{\mathcal{S}_1}$) still appears in the optimal learning rates (Equations (12) and (16)) and are essential for proving convergence (see Appendix A). These expressions also omit the influence of the smoothness constant, which affects the learning rate, as addressed in the convergence analysis.

### 3.3. Examining Definitions of Critical Batch Size

To formally link the GNS to CBS, we consolidate the optimal expected single-step improvements for each method by plugging their optimal learning rates (Equations (5), (12) and (16)) into their respective expected improvement bounds (Equations (4), (11) and (15)):

$$\Delta_k^*(B_k, \| \cdot \|_2) = \frac{\|\nabla\mathcal{L}(x_k)\|_2^2}{2} \cdot \left(1 + \frac{\mathcal{B}_{\ell_2}(x_k)}{B_k}\right)^{-1},$$

$$\Delta_k^*(B_k, \| \cdot \|_\infty) = \frac{\|\nabla\mathcal{L}(x_k)\|_1^2}{2d}\left(1 - \sqrt{\frac{\mathcal{B}_{\ell_1}(x_k)}{B_k}}\right)^2, \quad (18)$$

$$\Delta_k^*(B_k, \| \cdot \|_{\mathcal{S}_\infty}) = \frac{\|\nabla\mathcal{L}(X_k)\|_{\mathcal{S}_1}^2}{2r}\left(1 - \sqrt{\frac{\mathcal{B}_{\mathcal{S}_1}(x_k)}{B_k}}\right)^2.$$

These quantities increase monotonically with the batch size before eventually saturating, consistent with the empirical behavior observed in Shallue et al. (2019).

The CBS was originally defined as the "turning point" in batch size beyond which further increases yield diminishing returns in expected improvement. Following McCandlish et al. (2018), we define this as the batch size $B_{\text{CBS}}$ required to achieve a specific fraction $\kappa \in (0, 1)$ of the maximum possible deterministic improvement $\Delta_k^*(\infty, \| \cdot \|)$:

$$B_{\text{CBS},k}(\kappa, \|\cdot\|) = \inf\{B_k : \Delta_k^*(B_k, \|\cdot\|) \geq \kappa\Delta_k^*(\infty, \|\cdot\|)\}.$$

Solving this inequality for the specific forms in Equa-

tion (18) yields:

$$B_{\text{CBS},k}(\kappa, \| \cdot \|_2) = \frac{\kappa}{1 - \kappa}\mathcal{B}_{\ell_2}(x_k),$$

$$B_{\text{CBS},k}(\kappa, \| \cdot \|_\infty) = \left(\frac{1}{1 - \sqrt{\kappa}}\right)^2 \mathcal{B}_{\ell_1}(x_k), \quad (19)$$

$$B_{\text{CBS},k}(\kappa, \| \cdot \|_{\mathcal{S}_\infty}) = \left(\frac{1}{1 - \sqrt{\kappa}}\right)^2 \mathcal{B}_{\mathcal{S}_1}(x_k).$$

Irrespective of the specific choice of $\kappa$, the CBS is always a linear scaling of the respective GNS.

**Turning point definitions.** The constant $\kappa$ defines where on the saturation curve the "point of diminishing returns" occurs. Conceptually, this threshold separates two regimes: when $B_k \ll \mathcal{B}(x_k)$, optimization is noise-dominated and scaling the batch size yields a near-linear reduction in training steps; when $B_k \gg \mathcal{B}(x_k)$, the gradient error saturates, rendering additional samples computationally redundant.

While $\kappa = 1/2$ is standard for SGD (McCandlish et al., 2018), the saturation curves for non-Euclidean norms exhibit different geometric properties. As seen in Equation (18), the SGD improvement curve is strictly concave, whereas the curves for $\ell_\infty$ and $\mathcal{S}_1$ are initially convex, then concave. This implies that for non-Euclidean optimizers, there exists an inflection point where the rate of marginal gain peaks, motivating alternative definitions of $\kappa$ (see Appendix B). However, since our derivation relies on lower bounds, the theoretical constant relating CBS to GNS is loose. Therefore, in practice, we treat the GNS scaling as a tunable hyperparameter $\theta$ to empirically traverse the Pareto frontier between sample efficiency and steps.

## 4. Implementation

To translate our theoretical findings into a practical system, we introduce a variance estimation framework designed for distributed training. Our approach treats the local mini-batch gradients on each data-parallel rank as independent samples from the gradient distribution, enabling us to estimate population statistics without expensive auxiliary passes. This methodology extends the Euclidean variance estimation techniques from Lau et al. (2024a;b) to non-Euclidean GNS metrics required for signSGD and specSGD.

### 4.1. Noise Estimation

In a distributed data-parallel (DDP) setup with $R$ ranks, the global batch $\mathcal{S}_k$ of size $B_k$ is partitioned into disjoint local batches $\mathcal{S}_k^j$ of size $\frac{B_k}{R}$. Each rank $j$ computes a local mini-batch gradient $g_k^j = \frac{R}{B_k} \sum_{\xi \in \mathcal{S}_k^j} \nabla\ell(x_k, \xi)$. These local updates are aggregated to form the global gradient $g_k =$

$\frac{1}{R} \sum_{j=1}^{R} g_k^j$ via an `AllReduce` operation.[6]

We leverage the local mini-batch gradients $g_k^j$ to construct an unbiased estimator for the gradient noise. The estimators for the coordinate-wise variance $\hat{\sigma}_k$ and the row-wise covariance matrix $\hat{C}_{\text{row},k}$ are:

$$[\hat{\sigma}_k]_i^2 = \frac{B_k}{R-1} \left( \frac{1}{R} \sum_{j=1}^{R} [g_k^j]_i^2 - [g_k]_i^2 \right), \quad (20)$$

$$\hat{C}_{\text{row},k} = \frac{B_k}{R-1} \left( \frac{1}{R} \sum_{j=1}^{R} G_k^j (G_k^j)^\top - G_k G_k^\top \right). \quad (21)$$

The scaling factor $\frac{B_k}{R-1}$ applies two necessary corrections simultaneously: the Bessel correction $\frac{R}{R-1}$ converts the sample variance of the ranks into an unbiased population estimate, and the factor $\frac{B_k}{R}$ rescales the variance of the mini-batches to the variance of a single sample.

Using the global gradient ($g_k$ or $G_k$) as the signal estimate (following Bollapragada et al. (2018a)), we obtain the final computable GNS metrics:

$$\hat{\mathcal{B}}_{\ell_1} = \frac{\|\hat{\sigma}_k\|_1^2}{\|g_k\|_1^2} \quad \text{and} \quad \hat{\mathcal{B}}_{\mathcal{S}_1} = \frac{\|\hat{C}_{\text{row},k}^{1/2}\|_{\mathcal{S}_1}^2}{\|G_k\|_{\mathcal{S}_1}^2}. \quad (22)$$

All estimations are performed prior to gradient clipping.

**Implementation details.** For the $\ell_1$ GNS, each rank computes the element-wise square $(g_k^j)^2$ locally. In standard DDP, this requires storing one additional gradient copy per rank and performing an additional `AllReduce` to average the squares. In FSDP, this memory overhead is mitigated by instead performing a `ReduceScatter` on the squared gradients, effectively sharding the variance statistics consistent with the parameters.

For the $\mathcal{S}_1$ GNS, each rank computes the local Gram matrix $G_k^j (G_k^j)^\top \in \mathbb{R}^{m \times m}$. While FSDP can similarly `ReduceScatter` this term, computing the final nuclear norm $\|\hat{C}_{\text{row},k}^{1/2}\|_{\mathcal{S}_1}$ requires a subsequent `AllGather` to reconstruct the full covariance matrix on each rank. Because these computations are restricted to element-wise or block-wise statistics, the metric can be computed independently for each model component, ensuring full compatibility with tensor and pipeline parallelism. We provide a detailed wall-clock, communication, and memory cost analysis of the $\ell_1$ and $\mathcal{S}_1$ GNS estimators in Appendix E.

---

[6]In Fully-Sharded Data-Parallel (FSDP), the `AllReduce` is decomposed into `AllGather` and `ReduceScatter` primitives. Our variance computation hooks are inserted prior to the `ReduceScatter` to access the unsharded local gradients.

---

**Algorithm 1** Adaptive Batch Sizes (Non-Euclidean GNS)

1: **Input:** $\beta^N, \beta^M \in [0,1)$; update frequency $F$; initial warm-up $I$; total steps $K$; $\theta \in (0,1)$; learning rate sequence $\{\eta_k\}$; learning rate scaling $\omega_0 = 1$.
2: **for** $k = 0, ..., K-1$ **do**
3:     Compute $g_k$ and estimate $\hat{\sigma}_k$ using (20) or $\hat{C}_{\text{row},k}^{1/2}$ using (21).
4:     Compute $p_k = \arg\max_{\|p\| \leq 1} \langle g_k, p \rangle$.
5:     Update $x_{k+1} = x_k - \omega_k \eta_k p_k$.
6:     **if** $k \bmod F = 0$ **then**
7:         Update $N_k$ and $M_k$ using (23).
8:         **if** $k \geq I$ **then**
9:             Update $B_{k+1} = \max \left\{ \left\lceil \frac{N_k}{\theta^2 M_k} \right\rceil, B_k \right\}$.
10:            Update $\omega_{k+1} = \omega_k \sqrt{\frac{B_{k+1}}{B_k}}$.
11:         **else**
12:            Update $B_{k+1} = B_k$ and $\omega_{k+1} = \omega_k$.
13:         **end if**
14:     **end if**
15: **end for**

### 4.2. Adaptive Batch Size Heuristics

We implement several heuristics to stabilize our adaptive strategy (Algorithm 1). First, we apply *exponential moving averages* to the noise ($N_k$) and signal ($M_k$) components to mitigate the impact of stochastic fluctuations. For instance, for signSGD:

$$N_k = \beta^N N_{k-1} + (1 - \beta^N) \|\hat{\sigma}_k\|_1^2,$$
$$M_k = \beta^M M_{k-1} + (1 - \beta^M) \|g_k\|_1^2, \quad (23)$$

yielding the batch size $B_k = N_k / (\theta^2 M_k)$. Second, the noise ($N_k$) and signal ($M_k$) components are *updated periodically* to reduce computational overhead (Belias et al., 2025). Batch size increases are forced to occur after an initial warmup and grow monotonically to ensure scaling aligns with an improving signal-to-noise ratio while preventing reversals caused by transient noise. Third, to maximize the benefits of lower-variance gradient estimates, we *scale the learning rate proportional to the square root of the batch size* ($\eta_k \propto \sqrt{B_k}$) (Merrill et al., 2026). Finally, for composite optimizers that utilize specSGD and Muon for hidden layers and AdamW for 1D parameters, we compute the GNS solely from the 2D parameter groups, as these dominate the model's total parameter count.

## 5. Experiments

To evaluate the empirical effectiveness of our methodology, we conduct language and vision training experiments across a diverse set of optimizers, including signSGD, Signum, AdamW, specSGD and Muon. We compare our proposed adaptive strategy detailed in Algorithm 1 against constant

*Table 2.* Validation loss and steps reduction (%) for the 160M Llama 3 model trained for 3.2B tokens (10 seeds) over the C4 dataset. The adaptive batch size methods start from the optimal constant baseline ($B = 256$ for signSGD and $B = 64$ for all others). The steps reduction (%) represents the median percent reduction in steps to reach the baseline's minimum validation loss by Non-Euclidean GNS.

| OPTIMIZER | VALIDATION LOSS | | | | STEPS REDUCTION(%) |
| | B = 64 | B = 256 | EUCLIDEAN GNS | NON-EUCLIDEAN GNS | |
|---|---|---|---|---|---|
| SIGNSGD | $4.1390 \pm 0.0235$ | $3.9357 \pm 0.0057$ | $3.8906 \pm 0.0117$ | $3.8396 \pm 0.0012$ | 22.58 |
| SIGNUM | $3.3737 \pm 0.0045$ | $3.4460 \pm 0.0075$ | $3.3842 \pm 0.0055$ | $3.3707 \pm 0.0026$ | 66.61 |
| ADAMW | $3.2991 \pm 0.0054$ | $3.3228 \pm 0.0101$ | $3.3127 \pm 0.0101$ | $3.3031 \pm 0.0054$ | 67.13 |
| SPECSGD | $3.7489 \pm 0.0096$ | $3.9404 \pm 0.0074$ | $3.7436 \pm 0.0274$ | $3.7285 \pm 0.0094$ | 16.49 |
| MUON | $3.3041 \pm 0.0021$ | $3.3411 \pm 0.0019$ | $3.3181 \pm 0.0072$ | $3.3061 \pm 0.0027$ | 66.77 |

small and large batch size baselines. See Appendix C for further details regarding our experimental setup.

### 5.1. Language Workloads

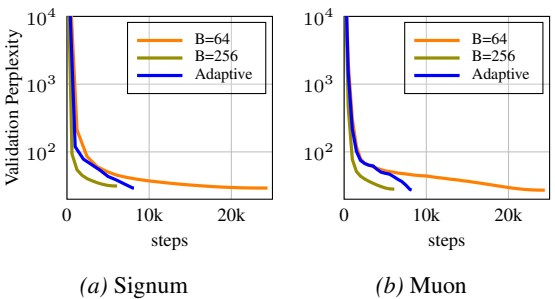

*(a)* Signum      *(b)* Muon

*Figure 2.* Comparison of constant batch sizes ($B = 64$ and $B = 256$) and our adaptive batch size method for the 160M Llama 3 model. For both Signum (Fig 2a) and Muon (Fig 2b), our adaptive strategy matches the final perplexity of the smaller batch size baseline while significantly reducing the total steps.

We train 160M and 1B parameter Llama 3 models (Grattafiori et al., 2024) on the C4 dataset (Raffel et al., 2020) using Chinchilla-optimal (Hoffmann et al., 2022) budgets of 3.2B and 22B tokens, respectively. Both configurations used a 2048 sequence length, trained via DDP on a single node with eight H100 GPUs (97GB memory). Our experimental results for the 160M and 1B models are summarized in Tables 2 and 3, respectively.

As illustrated in Figure 2, our proposed adaptive batch size strategy reduces the total steps for Signum and Muon while matching the validation perplexity of the constant small-batch baseline. Both signSGD and specSGD are able to achieve a significantly better loss than their constant batch size baselines. For the 160M Llama 3 model, the adaptive approach achieves up to a 66.77% reduction in steps required to reach the baseline minimum validation loss for Muon. For the 1B Llama 3 model, signSGD and Signum similarly demonstrate substantial efficiency gains of 31.84% and 12.11%, respectively, while AdamW does not achieve the baseline validation loss under the adaptive strategy. We provide additional ablation studies in Appendix D.1.

### 5.2. Vision Workloads

We train a SimpleViT (Beyer et al., 2022) on the Imagewoof (Howard, 2019) dataset. For each evaluated optimizer, we compare a constant batch ($B = 128$) baseline against our proposed adaptive batch size strategy, reporting performance from the best configuration found from our hyperparameter sweep. We also conduct evaluations on other architectures and datasets; see Appendix C.2 and D.2.

As shown in Table 4, for most optimizers, adaptive batching achieves comparable or lower validation loss while reducing the number of optimizer steps.

Overall, our results indicate that using $\ell_1$ and $\mathcal{S}_1$ GNS effectively reduces the number of steps on vision tasks without compromising validation performance.

## 6. Conclusion

In this work, we established the $\ell_1$ and $\mathcal{S}_1$ GNS as rigorous frameworks for adaptively selecting batch sizes for signSGD and specSGD. While this provides a robust foundation for generalized steepest descent, extending these principles to preconditioned and stateful optimizers remains an open challenge. For instance, how do we handle exponential moving averages of the gradients or second moments (e.g., in AdamW or Muon)? How do we unify different noise scales when different modules utilize different optimizers? Additionally, how do we account for more general Hessian structures in our GNS metrics?

Beyond batch size selection, the geometry-aware GNS can serve as a fundamental diagnostic tool for modern training stacks. By quantifying noise relative to the optimizer's update direction, this metric offers a precise lens for monitoring training stability, detecting distribution shifts, and guiding decisions in quantization or sparsity. We hope these signals motivate the development of next-generation training algorithms that explicitly leverage variance metrics and covariance adaptation.

*Table 3.* Validation loss and steps reduction (%) for the 1B Llama 3 model trained for 22B tokens (10 seeds) over the C4 dataset. The adaptive batch size method starts at $B = 64$, compared against the $B = 256$ constant-batch baseline. The steps reduction (%) indicates the percent reduction in steps to reach the baseline's validation loss.

| OPTIMIZER | VALIDATION LOSS | | STEPS REDUCTION (%) |
|---|---|---|---|
| | B = 256 | NON-EUCLIDEAN GNS | |
| SIGNSGD | $3.1417 \pm 0.0047$ | $2.9946 \pm 0.0201$ | 31.84 |
| SIGNUM | $2.8306 \pm 0.0051$ | $2.8354 \pm 0.0073$ | 12.11 |
| ADAMW | $2.7701 \pm 0.0037$ | $2.8023 \pm 0.0049$ | - |

*Table 4.* Validation loss and steps reduction (%) for the SimpleViT model trained (5 seeds) over ImageWoof dataset. The adaptive batch method starts at $B = 128$, compared against the $B = 128$ constant-batch baseline. The steps reduction (%) indicates the percent reduction in steps to reach the baseline's minimum validation loss. The $B = 512$ column is reported for reference. The Adaptive column uses the GNS measured in the dual norm matched to each optimizer's geometry ($\ell_2$ for SGD/MSGD/NormalizedSGD, $\ell_1$ for signSGD/Signum/AdamW, and $\mathcal{S}_1$ for Muon).

| OPTIMIZER | VALIDATION LOSS | | | STEPS REDUCTION(%) |
|---|---|---|---|---|
| | B = 128 | B = 512 | ADAPTIVE | |
| MSGD | $1.1966 \pm 0.0129$ | $1.4539 \pm 0.0118$ | $1.1849 \pm 0.0044$ | 4.14 |
| SGD | $1.4829 \pm 0.0187$ | $1.6473 \pm 0.0076$ | $1.3179 \pm 0.0239$ | 40.29 |
| NORMALIZEDSGD | $1.1979 \pm 0.0087$ | $1.5610 \pm 0.0065$ | $1.3286 \pm 0.0028$ | - |
| SIGNSGD | $1.2003 \pm 0.0087$ | $1.5153 \pm 0.0080$ | $1.1927 \pm 0.0064$ | 27.44 |
| SIGNUM | $1.2987 \pm 0.0217$ | $1.5109 \pm 0.0049$ | $1.2322 \pm 0.0029$ | 11.30 |
| ADAMW | $0.9750 \pm 0.0151$ | $1.0653 \pm 0.0051$ | $0.9338 \pm 0.0030$ | 10.86 |
| MUON | $0.6578 \pm 0.0218$ | $0.7156 \pm 0.0060$ | $0.6433 \pm 0.0057$ | 57.65 |

## Impact Statement

This paper presents work whose goal is to advance the field of machine learning. There are many potential societal consequences of our work, none of which we feel must be specifically highlighted here.

## Acknowledgements

We thank Raghu Bollapragada, Anna Cai, Jiaming Cui, Aaron Defazio, Runa Eschenhagen, Wei Feng, Chien-Chin Huang, Tsung-Hsien Lee, Gavin Zhang, and Iris Zhang for their discussions on the algorithm, support in experimentation, and detailed feedback on the manuscript. We also thank Adnan Aziz, Maxim Naumov, Sandeep Parab, Joel Pobar, and Chunqiang Tang for their managerial support of this work.

This work was supported by RBC Borealis through the RBC Borealis AI Global Fellowship Award, which was awarded to Hiroki Naganuma. This research was supported by grants from NVIDIA and utilized NVIDIA A100 GPUs provided through the NVIDIA Academic Grant Program. The authors also acknowledge the support of Compute Canada and Mila computing clusters for experimental resources.

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

# A. Theoretical Results

This section provides formal derivations for the theoretical bounds introduced in Section 3 and details the convergence analyses for signSGD and specSGD under the adaptive batch size strategy.

**Lemma A.1.** *Let $g_k$ be a gradient estimate such that $\mathbb{E}_k\left[g_k\right] = \nabla\mathcal{L}(x_k)$. Let $\|\cdot\|$ be a general norm and $\|\cdot\|_*$ be its associated dual norm. The stochastic steepest descent direction $p_k = \arg\max_{\|p\|\leq 1}\langle g_k, p\rangle$ satisfies:*

$$\mathbb{E}_k\left[\langle\nabla\mathcal{L}(x_k), p_k\rangle\right] \geq \|\nabla\mathcal{L}(x_k)\|_* - \mathbb{E}_k\left[\|\nabla\mathcal{L}(x_k) - g_k\|_*\right].$$

*Proof.* Let $\varepsilon_k = g_k - \nabla\mathcal{L}(x_k)$ be the gradient estimation error at iteration $k$. The inner product $\langle\nabla\mathcal{L}(x_k), p_k\rangle$ can be lower bounded as:

$$
\begin{aligned}
\langle\nabla\mathcal{L}(x_k), p_k\rangle &= \langle g_k - \varepsilon_k, p_k\rangle \\
&= \langle g_k, p_k\rangle - \langle\varepsilon_k, p_k\rangle \\
&= \|g_k\|_* - \langle\varepsilon_k, p_k\rangle \\
&\geq \|g_k\|_* - \|\varepsilon_k\|_*
\end{aligned}
\tag{24}
$$

where the third equality follows from the definition of the dual norm (since $p_k$ is chosen to maximize the inner product with $g_k$), and the final bound follows from Hölder's Inequality:

$$\langle\varepsilon_k, p_k\rangle \leq \|\varepsilon_k\|_*\|p_k\| = \|\varepsilon_k\|_*$$

(noting that $\|p_k\| \leq 1$).

Taking the expectation of the inequality (24) given $x_k$, we apply Jensen's inequality ($\mathbb{E}_k\left[\|g_k\|_*\right] \geq \|\mathbb{E}_k\left[g_k\right]\|_* = \|\nabla\mathcal{L}(x_k)\|_*$) to the first term to complete the proof.

$\square$

**Lemma A.2.** *Let $g_k = \frac{1}{B_k}\sum_{\xi\in\mathcal{S}_k}\nabla\ell(x_k;\xi) \in \mathbb{R}^d$ be a gradient estimate computed over a mini-batch $\mathcal{S}_k$ of $B_k$ i.i.d samples drawn from $\mathcal{D}$ at iteration $k$. Then the expected $\ell_1$ norm of the estimation error can be bounded as:*

$$\mathbb{E}_k\left[\|\nabla\mathcal{L}(x_k) - g_k\|_1\right] \leq \frac{\|\sigma_k\|_1}{\sqrt{B_k}},$$

*where $\sigma_k$ is the component-wise standard deviation for the single sample gradient $\nabla\ell(x_k;\xi)$ given $x_k$, i.e., for each component $i = 1, ..., d$:*

$$[\sigma_k]_i^2 = \mathbb{E}\left[\left([\nabla\ell(x_k;\xi) - \nabla\mathcal{L}(x_k)]_i\right)^2\right].$$

*Proof.* Let $[x]_i$ denote the $i$-th element of the vector $x \in \mathbb{R}^d$. Then:

$$
\begin{aligned}
\mathbb{E}_k\left[\|\nabla\mathcal{L}(x_k) - g_k\|_1\right] &= \sum_{i=1}^{d}\mathbb{E}_k\left[|[\nabla\mathcal{L}(x_k) - g_k]_i|\right] \\
&\leq \sum_{i=1}^{d}\sqrt{\mathbb{E}_k\left[\left([\nabla\mathcal{L}(x_k) - g_k]_i\right)^2\right]},
\end{aligned}
$$

where the final inequality follows from Jensen's inequality. Since $g_k$ is the average of $B_k$ i.i.d. samples, each with variance $[\sigma_k]_i^2$, we obtain:

$$\mathbb{E}_k\left[\|\nabla\mathcal{L}(x_k) - g_k\|_1\right] \leq \sum_{i=1}^{d}\sqrt{\frac{[\sigma_k]_i^2}{B_k}} = \frac{\|\sigma_k\|_1}{\sqrt{B_k}},$$

completing the proof.

$\square$

**Lemma A.3.** *Let $G_k = \frac{1}{B_k} \sum_{\xi \in \mathcal{S}_k} \nabla \ell(X_k; \xi)$ be a gradient estimate computed over a mini-batch $\mathcal{S}_k$ of $B_k$ i.i.d samples drawn from $\mathcal{D}$ at iteration $k$. Then the expected nuclear norm of the estimation error can be bounded as:*

$$\mathbb{E}_k \left[ \|\nabla \mathcal{L}(X_k) - G_k\|_{\mathcal{S}_1} \right] \leq \frac{\|C_{row,k}^{1/2}\|_{\mathcal{S}_1}}{\sqrt{B_k}},$$

*where $C_{row,k} = \mathbb{E}_k \left[ (\nabla \mathcal{L}(X_k) - \nabla \ell(X_k; \xi)) (\nabla \mathcal{L}(X_k) - \nabla \ell(X_k; \xi))^\top \right]$ is the row-wise aggregated covariance matrix.*

*Proof.* Let $\mathcal{E}_k = \nabla \mathcal{L}(X_k) - G_k \in \mathbb{R}^{m \times n}$ be the gradient estimate error. Then:

$$\mathbb{E}_k \left[ \|\mathcal{E}_k\|_{\mathcal{S}_1} \right] = \mathbb{E}_k \left[ \text{tr} \left( (\mathcal{E}_k \mathcal{E}_k^\top)^{1/2} \right) \right] \leq \text{tr} \left( \mathbb{E}_k \left[ \mathcal{E}_k \mathcal{E}_k^\top \right]^{1/2} \right) = \frac{\text{tr} \left( C_{row,k}^{1/2} \right)}{\sqrt{B_k}},$$

where the inequality follows from Jensen's inequality together with the concavity of the matrix square root over positive semi-definite matrices.

$\square$

**Theorem A.4.** *Assume the objective function $\mathcal{L}(x) = \mathbb{E}[\ell(x; \xi)] : \mathbb{R}^d \to \mathbb{R}$ is $\ell_\infty$ smooth, i.e.,*

$$\|\nabla \mathcal{L}(x) - \nabla \mathcal{L}(x')\|_1 \leq L_\infty \|x - x'\|_\infty \quad \forall x, x' \in \mathbb{R}^d,$$

*with $L_\infty > 0$ and bounded below with optimal value $\mathcal{L}^*$. Let $\{x_k\}$ be the sequence generated by the update rule $x_{k+1} = x_k - \eta_k \, \text{sign}(g_k)$ for $k \geq 0$, where $g_k = \frac{1}{B_k} \sum_{\xi \in \mathcal{S}_k} \nabla \ell(x_k; \xi)$ is a gradient estimate computed over a mini-batch of $B_k = \frac{\|\sigma_k\|_1^2}{\theta^2 \|\nabla \mathcal{L}(x_k)\|_1^2}$ i.i.d samples with $\theta \in (0, 1)$ and $\sigma_k$ the component-wise standard deviation for the single sample gradient $\nabla \ell(x_k; \xi)$ given $x_k$.*

1. *If the learning rate is chosen as $\eta_k = \eta = \frac{1}{\sqrt{L_\infty K}}$ for $K \geq 1$,*

$$\mathbb{E} \left[ \frac{1}{K} \sum_{k=0}^{K-1} \|\nabla \mathcal{L}(x_k)\|_1 \right] \leq \frac{\sqrt{L_\infty}}{(1 - \theta)\sqrt{K}} \left[ \mathcal{L}(x_0) - \mathcal{L}^* + \frac{1}{2} \right]. \tag{25}$$

2. *If the learning rate is chosen such that $\sum_{k=0}^\infty \eta_k = \infty$ and $\sum_{k=0}^\infty \eta_k^2 < \infty$,*

$$\liminf_{k \to \infty} \mathbb{E}[\|\nabla \mathcal{L}(x_k)\|_1] = 0. \tag{26}$$

3. *If the learning rate is chosen as $\eta_k = (1 - \theta)\frac{\|\nabla \mathcal{L}(x_k)\|_1}{L_\infty}$, then for any for $K \geq 1$,*

$$\mathbb{E} \left[ \frac{1}{K} \sum_{k=0}^{K-1} \|\nabla \mathcal{L}(x_k)\|_1^2 \right] \leq \frac{2 L_\infty}{(1 - \theta)^2 K} (\mathcal{L}(x_0) - \mathcal{L}^*). \tag{27}$$

*If the objective function is also $\mu_\infty$−strongly convex, i.e.,*

$$2\mu_\infty (\mathcal{L}(x) - \mathcal{L}^*) \leq \|\nabla \mathcal{L}(x)\|_1^2 \quad \forall x \in \mathbb{R}^d,$$

*with $\mu_\infty > 0$, then*

$$\mathbb{E}[\mathcal{L}(x_{k+1})] - \mathcal{L}^* \leq \left[ 1 - (1 - \theta)^2 \frac{\mu_\infty}{L_\infty} \right] (\mathbb{E}[\mathcal{L}(x_k)] - \mathcal{L}^*) \tag{28}$$

*and $\{x_k\}$ converges to a unique optimal point in expectation at a linear rate.*

*Proof.* Under the stated smoothness assumption, the change in the loss function in iteration $k$ can be bounded using Proposition 2 from Balles et al. (2020) as

$$\mathcal{L}(x_{k+1}) - \mathcal{L}(x_k) \leq -\eta_k \nabla \mathcal{L}(x_k)^T \operatorname{sign}(g_k) + \frac{L_\infty}{2} \|\eta_k \operatorname{sign}(g_k)\|_\infty^2 \leq -\eta_k \nabla \mathcal{L}(x_k)^T \operatorname{sign}(g_k) + \eta_k^2 \frac{L_\infty}{2}.$$

where the last inequality follows from $\|\operatorname{sign}(g_k)\|_\infty \leq 1$. Taking expectation of the above bound given $x_k$ and using Lemma A.1 and Lemma A.2,

$$\mathbb{E}_k \left[ \mathcal{L}(x_{k+1}) \right] - \mathcal{L}(x_k) \leq -\eta_k \|\nabla \mathcal{L}(x_k)\|_1 + \frac{\eta_k \|\sigma_k\|_1}{\sqrt{B_k}} + \eta_k^2 \frac{L_\infty}{2} = -\eta_k (1 - \theta) \|\nabla \mathcal{L}(x_k)\|_1 + \eta_k^2 \frac{L_\infty}{2}, \tag{29}$$

where the equality follows from the defined batch size.

Taking the total expectation of Equation (29) with respect to the entire trajectory of updates, we obtain a telescopic sum for $k = 0, 1, 2, \ldots, K - 1$,

$$\begin{aligned}
\mathcal{L}(x_0) - \mathcal{L}^* &\geq \mathcal{L}(x_0) - \mathbb{E}\left[ \mathcal{L}(x_K) \right] \\
&= \sum_{k=0}^{K-1} \mathbb{E}\left[ \mathcal{L}(x_k) - \mathcal{L}(x_{k+1}) \right] \\
&\geq (1 - \theta) \left[ \sum_{k=0}^{K-1} \eta_k \|\nabla \mathcal{L}(x_k)\|_1 - \frac{L_\infty}{2} \sum_{k=0}^{K-1} \eta_k^2 \right],
\end{aligned}$$

where rearranging the in above using $1 - \theta > 0$ yields,

$$\mathbb{E}\left[ \sum_{k=0}^{K-1} \eta_k \|\nabla \mathcal{L}(x_k)\|_1 \right] \leq \frac{1}{(1 - \theta)} \left[ \mathcal{L}(x_0) - \mathcal{L}^* + \frac{L_\infty}{2} \sum_{k=0}^{K-1} \eta_k^2 \right]. \tag{30}$$

Substituting the learning rate as $\eta_k = (1 - \theta) \frac{\|\nabla \mathcal{L}_\infty\|_1}{L_\infty}$ Substituting the learning rate as $\eta_k = \eta = \frac{1}{\sqrt{L_\infty K}}$ in Equation (30) and multiplying by $\frac{1}{\sqrt{K}}$ yields Equation (25).

If the learning rate satisfies $\sum_{k=0}^{\infty} \eta_k = \infty$ and $\sum_{k=0}^{\infty} \eta_k^2 < \infty$, we can infer that $\lim_{K \to \infty} \sum_{k=0}^{K-1} \eta_k \mathbb{E}\left[ \|\nabla \mathcal{L}(x_k)\|_1 \right] < \infty$ and Equation (26) follows.

If the learning rate is chosen as $\eta_k = \frac{(1-\theta)\|\nabla \mathcal{L}(x_k)\|_1}{L_\infty}$ (now dependent on the trajectory of updates), substituting in Equation (29) yields,

$$\mathbb{E}_k \left[ \mathcal{L}(x_{k+1}) \right] - \mathcal{L}(x_k) \leq -\frac{(1 - \theta)^2 \|\nabla \mathcal{L}(x_k)\|_1^2}{2 L_\infty}, \tag{31}$$

where taking the total expectation and performing a telescopic sum as done for Equation (30) results in Equation (27).

If $\mathcal{L}(x)$ is also strongly convex, from Equation (31) as $1 - \theta > 0$,

$$\mathbb{E}_k \left[ \mathcal{L}(x_{k+1}) \right] - \mathcal{L}(x_k) \leq -\frac{(1 - \theta)^2 \|\nabla \mathcal{L}(x_k)\|_1^2}{2 L_\infty} \leq -\frac{(1 - \theta)^2 \mu_\infty}{L_\infty} \left( \mathcal{L}(x_k) - \mathcal{L}^* \right).$$

Taking total expectation of the above and rearranging the inequality yields Equation (28). $\qquad \square$

**Theorem A.5.** *Assume the objective function $\mathcal{L}(X) = \mathbb{E}\left[ \ell(X; \xi) \right] : \mathbb{R}^{m \times n} \to \mathbb{R}$ is smooth with respect to the spectral norm ($\| \cdot \|_{\mathcal{S}_\infty}$), i.e.,*

$$\|\nabla \mathcal{L}(X) - \nabla \mathcal{L}(X')\|_{\mathcal{S}_\infty} \leq L_{\mathcal{S}_\infty} \|X - X'\|_{\mathcal{S}_1} \quad \forall X, X' \in \mathbb{R}^{m \times n},$$

*where $\| \cdot \|_{\mathcal{S}_1}$ is the nuclear norm, with $L_{\mathcal{S}_\infty} > 0$ and bounded below by the optimal value $\mathcal{L}^*$. Let $\{X_k\}$ be the sequence generated by the update rule $X_{k+1} = X_k - \eta_k \operatorname{matsign}(G_k)$, where $G_k = \frac{1}{B_k} \sum_{\xi \in \mathcal{S}_k} \nabla \ell(X_k; \xi)$ is a gradient estimate computed over a mini-batch of $B_k = \frac{\|C_{row,k}^{1/2}\|_{\mathcal{S}_1}^2}{\theta^2 \|\nabla \mathcal{L}(X_k)\|_{\mathcal{S}_1}^2}$ i.i.d samples and $C_{row,k} = \mathbb{E}_k \left[ (\nabla \mathcal{L}(X_k) - \nabla \ell(X_k; \xi)) (\nabla \mathcal{L}(X_k) - \nabla \ell(X_k; \xi))^\top \right]$ is the row-wise aggregated covariance matrix.*

1. *If the learning rate is chosen as $\eta_k = \eta = \frac{1}{\sqrt{L_{\mathcal{S}_\infty} K}}$ for $K \geq 1$,*

$$\mathbb{E}\left[\frac{1}{K}\sum_{k=0}^{K-1}\|\nabla\mathcal{L}(x_k)\|_{\mathcal{S}_1}\right] \leq \frac{\sqrt{L_{\mathcal{S}_\infty}}}{(1-\theta)\sqrt{K}}\left[\mathcal{L}(x_0) - \mathcal{L}^* + \frac{1}{2}\right]. \tag{32}$$

2. *If the learning rate is chosen such that $\sum_{k=0}^{\infty}\eta_k = \infty$ and $\sum_{k=0}^{\infty}\eta_k^2 < \infty$,*

$$\liminf_{k\to\infty}\ \mathbb{E}[\|\nabla\mathcal{L}(X_k)\|_{\mathcal{S}_\infty}] = 0. \tag{33}$$

3. *If the learning rate is chosen as $\eta_k = \frac{(1-\theta)\|\nabla\mathcal{L}(X_k)\|_{\mathcal{S}_1}}{L_{\mathcal{S}_\infty}}$, then for $K \geq 1$,*

$$\mathbb{E}\left[\frac{1}{K}\sum_{k=0}^{K-1}\|\nabla\mathcal{L}(X_k)\|_{\mathcal{S}_1}^2\right] \leq \frac{2L_{\mathcal{S}_\infty}}{(1-\theta)^2 K}(\mathcal{L}(X_0) - \mathcal{L}^*). \tag{34}$$

*If the objective function is also $\mu_{\mathcal{S}_\infty}-$strongly convex, i.e.,*

$$2\mu_{\mathcal{S}_\infty}\left(\mathcal{L}(X) - \mathcal{L}^*\right) \leq \|\nabla\mathcal{L}(X)\|_{\mathcal{S}_1}^2 \quad \forall X \in \mathbb{R}^{m\times n},$$

*with $\mu_{\mathcal{S}_\infty} > 0$, then*

$$\mathbb{E}\left[\mathcal{L}(X_{k+1})\right] - \mathcal{L}^* \leq \left[1 - (1-\theta)^2\frac{\mu_{\mathcal{S}_\infty}}{L_{\mathcal{S}_\infty}}\right](\mathbb{E}\left[\mathcal{L}(X_k)\right] - \mathcal{L}^*) \tag{35}$$

*and $\{X_k\}$ converges to a unique optimal point in expectation at a linear rate.*

*Proof.* Under the stated smoothness assumption, from Carlson et al. (2015b), the change in the loss function in iteration $k$ can be bounded as

$$\mathcal{L}(X_{k+1}) - \mathcal{L}(X_k) \leq -\eta_k\langle\nabla\mathcal{L}(X_k), \mathrm{matsign}(G_k)\rangle + \frac{L_{\mathcal{S}_\infty}}{2}\|\eta_k\,\mathrm{matsign}(G_k)\|_{\mathcal{S}_\infty}^2$$

$$= -\eta_k\langle\nabla\mathcal{L}(X_k), \mathrm{matsign}(G_k)\rangle + \eta_k^2\frac{L_{\mathcal{S}_\infty}}{2},$$

where the equality follows from $\|\mathrm{matsign}(G_k)\|_{\mathcal{S}_\infty} = 1$. Taking expectation of the above bound given $x_k$ and using Lemma A.1 and Lemma A.3,

$$\mathbb{E}_k\left[\mathcal{L}(X_{k+1})\right] - \mathcal{L}(X_k) \leq -\eta_k\|\nabla\mathcal{L}(X_k)\|_{\mathcal{S}_1} + \eta_k\frac{\|C_{\mathrm{row},k}^{1/2}\|_{\mathcal{S}_1}}{\sqrt{B_k}} + \eta_k^2\frac{L_{\mathcal{S}_\infty}}{2} = -\eta_k(1-\theta)\,\|\nabla\mathcal{L}(x_k)\|_{\mathcal{S}_1} + \eta_k^2\frac{L_{\mathcal{S}_\infty}}{2}, \tag{36}$$

where the equality follows from the defined batch size.

Following the same procedure as Theorem A.4 of taking the total expectation of Equation (36) with respect to the entire trajectory of updates, taking the telescopic sum for $k = 0, 1, 2, \ldots, K-1$ and rearranging,

$$\mathbb{E}\left[\sum_{k=0}^{K-1}\eta_k\,\|\nabla\mathcal{L}(X_k)\|_{\mathcal{S}_1}\right] \leq \frac{1}{(1-\theta)}\left[\mathcal{L}(X_0) - \mathcal{L}^* + \frac{L_{\mathcal{S}_\infty}}{2}\sum_{k=0}^{K-1}\eta_k^2\right]. \tag{37}$$

Substituting the learning rate as $\eta_k = \eta = \frac{1}{\sqrt{L_{\mathcal{S}_\infty} K}}$ in Equation (37) and multiplying by $\frac{1}{\sqrt{K}}$ yields Equation (32).

If the learning rate satisfies $\sum_{k=0}^{\infty}\eta_k = \infty$ and $\sum_{k=0}^{\infty}\eta_k^2 < \infty$, we can infer that $\lim_{K\to\infty}\sum_{k=0}^{K-1}\eta_k\mathbb{E}\left[\|\nabla\mathcal{L}(X_k)\|_{\mathcal{S}_1}\right] < \infty$ and Equation (33) follows.

If the learning rate is chosen as $\eta_k = \frac{(1-\theta)\|\nabla\mathcal{L}(X_k)\|_{\mathcal{S}_1}}{L_{\mathcal{S}_\infty}}$ (now dependent on the trajectory of updates), substituting in Equation (36) yields,

$$\mathbb{E}_k\left[\mathcal{L}(X_{k+1})\right] - \mathcal{L}(X_k) \leq -\frac{(1-\theta)^2\|\nabla\mathcal{L}(X_k)\|_{\mathcal{S}_1}^2}{2L_{\mathcal{S}_\infty}}, \tag{38}$$

where taking the total expectation and performing a telescopic sum as done for Equation (37) results in Equation (34).

If $\mathcal{L}(X)$ is also strongly convex, from Equation (38) as $1 - \theta > 0$,

$$\mathbb{E}_k\left[\mathcal{L}(X_{k+1})\right] - \mathcal{L}(X_k) \leq -\frac{(1-\theta)^2\|\nabla\mathcal{L}(X_k)\|_{\mathcal{S}_1}^2}{2L_{\mathcal{S}_\infty}} \leq -\frac{(1-\theta)^2\mu_{\mathcal{S}_\infty}}{L_{\mathcal{S}_\infty}}\left(\mathcal{L}(X_k) - \mathcal{L}^*\right)$$

Taking total expectation of the above and rearranging the inequality yields Equation (35).

$\square$

## B. Alternative Definitions of Critical Batch Size

This appendix formalizes several common "turning-point" definitions of CBS. For signSGD and specSGD, these definitions differ only by constant factors while using the same intrinsic scale of GNS.

Recall from Section 3.3, Equation (18), for the non-Euclidean optimizers analyzed, the optimal single-step expected loss improvement takes the form:

$$\Delta^*(B) = \Delta^*_{\max}\left(1 - \sqrt{\frac{\mathcal{B}}{B}}\right)^2, \quad \text{for } B > \mathcal{B}, \tag{39}$$

where $\Delta^*_{\max} = \lim_{B\to\infty}\Delta^*(B)$ is the deterministic limit and $\mathcal{B}$ is the geometry-dependent GNS:

$$\mathcal{B}_{\ell_1} = \left(\frac{\|\sigma\|_1}{\|g\|_1}\right)^2 \qquad\qquad \text{(signSGD, } \ell_\infty \text{ geometry)}, \tag{40}$$

$$\mathcal{B}_{\mathcal{S}_1} = \left(\frac{\|C_{\text{row},k}^{1/2}\|_{\mathcal{S}_1}}{\|G\|_{\mathcal{S}_1}}\right)^2 \qquad\qquad \text{(specSGD, Schatten-}\infty \text{ norm)}, \tag{41}$$

for our two non-Euclidean cases. Equation (39) arises by factoring the constants in the optimized one-step expressions (e.g., $\|\nabla\mathcal{L}(x_k)\|_1$) that do not affect the location of turning points in $B$.

**Definition B.1** (Fraction-of-Maximum CBS). The fraction-of-maximum CBS is defined as the batch size $B_{\text{CBS}}(\kappa)$ required to achieve a fraction $\kappa \in (0,1)$ of the maximum possible improvement:

$$B_{\text{CBS}}(\kappa) := \inf\left\{B : \Delta^*(B) \geq \kappa\Delta^*_{\max}\right\}. \tag{42}$$

**Derivation.** Solving $\left(1 - \sqrt{\mathcal{B}/B}\right)^2 = \kappa$ for $B$ yields:

$$B_{\text{CBS}}(\kappa) = \left(\frac{1}{1 - \sqrt{\kappa}}\right)^2 \mathcal{B}. \tag{43}$$

This confirms that the CBS is a constant scaling of the GNS.

**Connection to the $\theta$-parameterization.** If we select $B = \theta^{-2}\mathcal{B}$, as specified in Equation (7), then

$$\frac{\Delta^*(B)}{\Delta^*(\infty)} = (1-\theta)^2.$$

Thus, choosing $\theta$ is equivalent to choosing the fraction $\kappa = (1-\theta)^2$ in Equations (42) and (43), recovering $B = \theta^{-2}\mathcal{B}$.

**Definition B.2** (Inflection Point CBS). The inflection point CBS is defined as the batch size $B_{\text{infl}}$ corresponding to the inflection point of the improvement curve, where the rate of marginal gain begins to decrease:

$$B_{\text{infl}} := \left\{ B : \frac{d^2}{dB^2} \Delta^*(B) = 0 \right\}. \tag{44}$$

**Derivation.** Let $f(B) = (1 - \sqrt{\mathcal{B}/B})^2$. Differentiating twice with respect to $B$ yields:

$$f'(B) = \left(1 - \sqrt{\frac{\mathcal{B}}{B}}\right) \cdot \left(\sqrt{\mathcal{B}}\, B^{-3/2}\right), \tag{45}$$

$$f''(B) = -\frac{3}{2}\sqrt{\mathcal{B}}\, B^{-5/2} + 2\mathcal{B}\, B^{-3}. \tag{46}$$

Setting $f''(B) = 0$ and solving for $B$ gives

$$B_{\text{curv}} = \frac{16}{9}\mathcal{B}. \tag{47}$$

**Definition B.3** (Maximum Efficiency CBS). The maximum efficiency CBS is defined as the batch size $B_{\text{eff}}$ that maximizes the improvement per sample (computational efficiency):

$$B_{\text{eff}} := \arg\max_{B \geq 1} \frac{\Delta^*(B)}{B}. \tag{48}$$

**Derivation.** Let $h(B) = \frac{1}{B}(1 - \sqrt{\mathcal{B}/B})^2$ and $t = \sqrt{\mathcal{B}/B}$. Then we maximize $t^2(1-t)^2$ for $t \in (0, 1)$. The stationarity condition yields $t = 1/2$. Substituting back to $B$:

$$B_{\text{eff}} = 4\mathcal{B}. \tag{49}$$

**Instantiations for signSGD and specSGD.** Plugging Equation (40) into Equations (47) and (49) yields:

$$B_{\text{inf}}^{\text{sign}} = \frac{16}{9}\left(\frac{\|\sigma\|_1}{\|g\|_1}\right)^2, \qquad\qquad B_{\text{eff}}^{\text{sign}} = 4\left(\frac{\|\sigma\|_1}{\|g\|_1}\right)^2,$$

$$B_{\text{inf}}^{\text{spec}} = \frac{16}{9}\left(\frac{\|C_{\text{row},k}^{1/2}\|_{\mathcal{S}_1}}{\|G\|_{\mathcal{S}_1}}\right)^2, \qquad\qquad B_{\text{eff}}^{\text{spec}} = 4\left(\frac{\|C_{\text{row},k}^{1/2}\|_{\mathcal{S}_1}}{\|G\|_{\mathcal{S}_1}}\right)^2.$$

**Comparison with Euclidean SGD.** From Equation (18) and (McCandlish et al., 2018), SGD follows a saturation curve of the form $\Delta^*(B) \propto (1 + \mathcal{B}_{\ell_2}/B)^{-1}$. This function is strictly concave for $B > 0$, meaning it lacks an inflection point. Furthermore, the efficiency ratio $\Delta^*(B)/B$ is strictly decreasing, meaning the maximum efficiency occurs at $B = 1$. Thus, the definitions for $B_{\text{infl}}$ and $B_{\text{eff}}$ are ill-defined for standard SGD. This necessitates the use of the Fraction-of-Maximum definition (commonly with $\kappa = 0.5$, yielding $B = \mathcal{B}_{\ell_2}$) in the Euclidean setting. In contrast, the non-Euclidean geometries of signSGD and specSGD induce an initial convex phase in optimization, making $B_{\text{infl}}$ and $B_{\text{eff}}$ valid alternate targets. All three definitions, however, result in $B \propto \mathcal{B}$. In our experiments, we found the tuned value of $\theta$ to align closely with the maximum efficiency CBS for language models.

## C. Experimental Setup

### C.1. Language Models

For our language model experimental runs, we utilize a learning rate schedule with a linear warm-up for the first 15% of the total sample budget, followed by a cosine decay schedule. For the adaptive approach, we maintain a constant initial batch period ($I$) that matches the learning rate warm-up duration, measure the GNS every 100 iterations ($F = 100$), and fix the scaling coefficient at $\theta = 0.6$ for all configurations. Table 5 summarizes the search space for hyperparameter tuning.

### C.2. Vision Models

We run two image classification workloads that cover common vision architectures. First, we train SimpleViT (Beyer et al., 2022) over the Imagewoof dataset (Howard, 2019) which isolates the transformer setting in a vision task. Second, we train ResNet-18 over the CIFAR-10 dataset to cover CNNs with residual connections.

*Table 5.* Summary of hyperparameter search space for language modeling experiments.

| METHOD | HYPERPARAMETER | PARAMETER SET |
|---|---|---|
| ALGORITHM 1 | $(\beta^N, \beta^M)$ 
 $\theta$ 
 $F$ | $(0.9, 0.9)$ 
 $0.6$ 
 100 ITERATIONS |
| ALL METHODS | LEARNING RATE SCHEDULE 
 WARMUP FRACTION % 
 LEARNING RATE MIN FRACTION % 
 DECOUPLED WEIGHT DECAY 
 LEARNING RATE | COSINE 
 15% 
 0.0 
 0.1 
 $\{1e^{-4}, 2e^{-4}, 4e^{-4}, 6e^{-4}, 8e^{-4},$ 
 $1e^{-3}, 2e^{-3}, 4e^{-3}, 8e^{-3}, 1e^{-2}\}$ |
| SIGNUM | EMA PARAMETER $\beta$ | $\{0.7, 0.8, 0.85, 0.875, 0.9, 0.925$ 
 $0.95, 0.975, 0.9875, 0.999\}$ |
| ADAMW | $\beta_1$ 

 $\beta_2$ 

 $\epsilon$ | $\{0.7, 0.8, 0.85, 0.875, 0.9, 0.925$ 
 $0.95, 0.975, 0.9875, 0.999\}$ 
 $\{0.7, 0.8, 0.85, 0.875, 0.9, 0.925$ 
 $0.95, 0.975, 0.9875, 0.999\}$ 
 $1e^{-8}$ |
| MUON | NEWTON SCHULZ ITERATIONS 
 MOMENTUM $\beta$ | 5 
 $\{0.7, 0.8, 0.85, 0.875, 0.9, 0.925$ 
 $0.95, 0.975, 0.9875, 0.999\}$ |

Across both workloads, we compare constant batch training to the adaptive strategy in Algorithm 1. For ResNet-18 over the CIFAR-10, we train for 100 epochs, and for SimpleViT over the Imagewoof we use a constant batch baseline of $B = 128$. For adaptive batching, we fix $(\beta^N, \beta^M) = (0.9, 0.9)$ and sweep the tolerance parameter $\theta$. The batch size is updated periodically with period $F$ set to 1 epoch for ResNet-18 over the CIFAR-10 and 3 epochs for SimpleViT over the Imagewoof. Tables 6 and 7 summarize the hyperparameter search spaces.

## D. Additional Experimental Results

In this section, we provide additional numerical experiments to support the empirical claims made in the main body.

### D.1. Language Models

This section presents extended results for the 160M Llama 3 model trained using signSGD. Table 8 evaluates our adaptive approach against baseline strategies utilizing constant batch sizes. The adaptive batch size strategy consistently outperforms constant baselines, reaching lower validation loss in significantly fewer training steps. The results also indicate a clear performance gain when scaling the learning rate in tandem with the batch size.

Table 9 compares the performance of the adaptive strategy when using the $\ell_1$ and $\ell_2$ GNS metrics. For $\ell_2$, $\theta = 0.3$ is used after tuning. The results demonstrate the adaptive strategy via $\ell_1$ GNS yields better results.

To further assess the robustness of our approach, Figure 3 visualizes training trajectories across multiple seeds for the adaptive strategy (starting at a batch size of 64) and Table 10 provides sensitivity results for $\theta$. While individual seeds produce unique batch size paths in Figure 3, based on their specific gradient noise profiles, all runs yield better performance than the constant batch size baselines. This consistency demonstrates that the proposed method effectively adapts the training schedule to the observed data distribution. In Table 10, one can see that even with the change in results over different values for $\theta$, the result loss form adaptive strategy via $\ell_1$ GNS is still better than the best result via $\ell_2$ GNS.

### D.2. Vision Models

This section presents results for the ResNet-18 model trained over the CIFAR-10, comparing the constant batch size baseline at $(B = 256)$ and $(B = 512)$ to the proposed adaptive strategy across optimizers, summarized in Table 11. Compared to the baseline $(B = 256)$, adaptive batching maintains or improves the validation accuracy for five of the six optimizers, with

*Table 6.* Summary of hyperparameter search space used for training SimpleViT over Imagewoof dataset.

| METHOD | HYPERPARAMETER | PARAMETER SET |
|---|---|---|
| ALGORITHM 1 | $(\beta^N, \beta^M)$ 
 $\theta$ 
 $F$ | $(0.9, 0.9)$ 
 $\{0.25, 0.5, 1, 2\}$ 
 3 EPOCHS |
| ALL METHODS | EPOCHS 
 MOMENTUM 
 LEARNING RATE SCHEDULE 
 WARMUP FRACTION % 
 LEARNING RATE MIN VALUE 
 DECOUPLED WEIGHT DECAY | 100 
 $\{0.925, 0.95, 0.975, 0.9875, 0.999\}$ 
 COSINE 
 4% 
 $1e^{-6}$ 
 0.0001 |
| MSGD / SGD | LEARNING RATE 
 MOMENTUM | $\{0.005, 0.007, 0.01, 0.023, 0.05, 0.07, 0.1, 0.2\}$ 
 $0.925, 0.95, 0.975, 0.9875, 0.999$ |
| SIGNSGD | LEARNING RATE 
 EPOCHS | $\{0.001, 0.005, 0.007, 0.01, 0.05\}$ 
 200 |
| SIGNUM | LEARNING RATE 
 EMA PARAMETER $\beta$ | $\{1e^{-4}, 5e^{-4}, 1e^{-3}, 5e^{-3}, 1e^{-2}\}$ 
 $\{0.7, 0.8, 0.85, 0.875, 0.9, 0.925, 0.95, 0.975, 0.9875, 0.999\}$ |
| ADAMW | LEARNING RATE 
 $\beta_1$ 
 $\beta_2$ 
 $\epsilon$ | $\{1e^{-4}, 3e^{-4}, 6e^{-4}, 1e^{-3}, 3e^{-3}, 6e^{-3}, 1e^{-2}\}$ 
 $\{0.925, 0.95, 0.975, 0.9875, 0.999\}$ 
 0.999 
 $1e^{-8}$ |
| MUON | LEARNING RATE 
 MOMENTUM $\beta$ | $\{6e^{-4}, 1e^{-3}, 3e^{-3}, 6e^{-3}, 1e^{-2}\}$ 
 0.95 |

MSGD being the only exception. For the optimizers that reach the baseline threshold, adaptive batching reduces the number of optimization steps by 22.02–72.14%. The reduction is consistent for momentum SGD and sign-based methods, and it is largest for Muon, which also attains the best accuracy among the listed optimizers.

*Table 7.* Summary of hyperparameter search spaces for training ResNet-18 over CIFAR-10 dataset.

| METHOD | HYPERPARAMETER | PARAMETER SET |
|---|---|---|
| ALGORITHM 1 | $(\beta^N, \beta^M)$ 
 $\theta$ 
 $F$ 
 INITIAL BATCH SIZE | $(0.9, 0.9)$ 
 $\{0.125, 0.25, 0.5, 1, 2.0, 4.0\}$ 
 1 EPOCH 
 $\{64, 256\}$ |
| ALL METHODS | EPOCHS 
 LEARNING RATE SCHEDULE 
 WARMUP FRACTION % 
 LEARNING RATE MIN VALUE 
 DECOUPLED WEIGHT DECAY | 100 
 COSINE 
 5% 
 $1e^{-6}$ 
 0.0 |
| MSGD/SGD | LEARNING RATE 
 MOMENTUM | $\{0.1, 0.2, 0.3, 0.5, 0.75, 1.0\}$ 
 $\{0.9, 0.925, 0.95, 0.975\}$ |
| SIGNSGD | LEARNING RATE | $\{1e^{-4}, 1e^{-3}, 1e^{-2}\}$ |
| SIGNUM | LEARNING RATE 
 EMA PARAMETER $\beta$ | $\{1e^{-4}, 1e^{-3}, 1e^{-2}, 1e^{-1}\}$ 
 $\{0.9, 0.925, 0.95, 0.95\}$ |
| ADAMW | LEARNING RATE 
 $\beta_1$ 
 $\beta_2$ 
 $\epsilon$ | $\{1e^{-4}, 3e^{-4}, 1e^{-3}, 3e^{-3}, 1e^{-2}, 3e^{-2}, 1e^{-1}\}$ 
 0.9 
 0.999 
 $1e^{-8}$ |
| MUON | LEARNING RATE 
 MOMENTUM $\beta$ | $\{1e^{-4}, 1e^{-3}, 1e^{-2}, 1e^{-1}\}$ 
 $\{0.9, 0.925, 0.95, 0.95\}$ |

*Table 8.* Comparison of constant versus adaptive batch sizes for a 160M Llama 3 trained for 3.2B tokens (10 seeds) over the C4 dataset using signSGD. Results show that adaptive batch sizes achieve improved training efficiency over constant-batch baselines and better validation loss with learning rate scaling.

| BATCH SIZE METHOD | INITIAL BATCH SIZE | LEARNING RATE SCALING | VALIDATION LOSS | MEDIAN TRAINING STEPS |
|---|---|---|---|---|
| CONSTANT | 64 | - | $4.1390 \pm 0.0235$ | 24415 |
| | 128 | - | $3.9963 \pm 0.0098$ | 8426 |
| | 256 | - | $3.9357 \pm 0.0057$ | 6103 |
| ADAPTIVE | 64 | FALSE | $3.6857 \pm 0.0085$ | 9285 |
| | | TRUE | $\mathbf{3.6817} \pm 0.0094$ | 9950 |
| | 128 | FALSE | $3.7082 \pm 0.0068$ | 7617 |
| | | TRUE | $\mathbf{3.7029} \pm 0.0094$ | 7826 |
| | 256 | FALSE | $3.8539 \pm 0.0039$ | 5007 |
| | | TRUE | $\mathbf{3.8396} \pm 0.0012$ | 5274 |

*Table 9.* Comparison of adaptive batch sizes using $\ell_1$ and $\ell_2$ GNS with signSGD for 160M Llama 3 trained for 3.2B tokens (10 seeds) over the C4 dataset. Results show that adaptive batch sizes using $\ell_1$ GNS, which aligns with the optimizer geometry, performs better.

| GNS | INITIAL BATCH SIZE | VALIDATION LOSS | MEDIAN TRAINING STEPS |
|---|---|---|---|
| $\mathcal{B}_{\ell_2}$ | 64 | $3.7431 \pm 0.0402$ | 10798 |
| | 128 | $3.7752 \pm 0.0088$ | 8869 |
| | 256 | $3.8906 \pm 0.0117$ | 5574 |
| $\mathcal{B}_{\ell_1}$ | 64 | $\mathbf{3.6817} \pm 0.0094$ | 9950 |
| | 128 | $\mathbf{3.7029} \pm 0.0094$ | 7826 |
| | 256 | $\mathbf{3.8396} \pm 0.0012$ | 5274 |

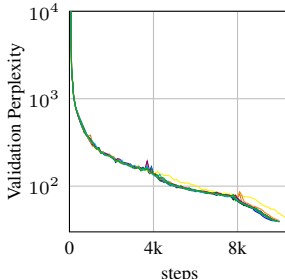 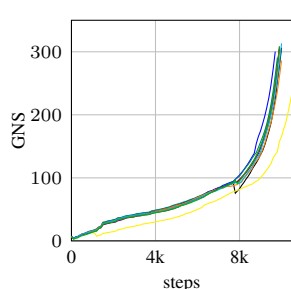 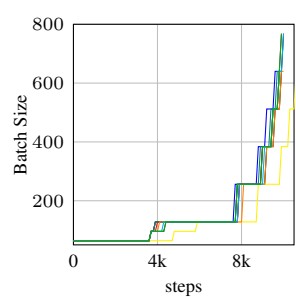 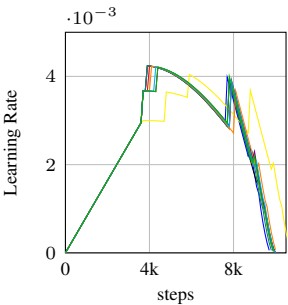

*Figure 3.* Validation perplexity, *exponential moving average* of GNS, batch size and learning rate for 10 seeds for the adaptive strategy starting with an initial batch size of 64 using signSGD for 160M Llama 3 trained for 3.2B tokens over the C4 dataset. The stability of the validation perplexity across multiple seeds underscores the reliability of our proposed method.

*Table 10.* Comparison of adaptive batch sizes using $\ell_1$ GNS with signSGD for 160M Llama 3 trained for 3.2B tokens (10 seeds) over the C4 dataset starting at batch size 64 over multiple values of $\theta$. Results show low sensitivity to $\theta$ and the results still performing better than $\ell_2$ GNS (Table 9).

| INITIAL BATCH SIZE | $\theta$ | VALIDATION LOSS | MEDIAN TRAINING STEPS |
|---|---|---|---|
| 64 | 0.5 | $3.6909 \pm 0.0097$ | 7650 |
| | 0.6 | $3.6817 \pm 0.0094$ | 9950 |
| | 0.7 | $3.7292 \pm 0.0092$ | 12374 |

*Table 11.* Validation accuracy and step reduction (%) for the ResNet-18 model trained (5 seeds) over the CIFAR-10 dataset. The steps reduction (%) indicates the percentage reduction in steps required to reach the best validation accuracy of the $B = 256$ baseline. The $B = 512$ column is reported for reference. The Adaptive column uses the GNS measured in the dual norm matched to each optimizer's geometry ($\ell_2$ for SGD/MSGD/, $\ell_1$ for signSGD/Signum/AdamW, and $\mathcal{S}_1$ for Muon).

| OPTIMIZER | VALIDATION ACCURACY | | | STEPS REDUCTION(%) |
|---|---|---|---|---|
| | B = 256 | B = 512 | ADAPTIVE | |
| MSGD | $93.5680 \pm 0.1669$ | $93.4080 \pm 0.2810$ | $93.3060 \pm 0.1270$ | - |
| SGD | $92.5700 \pm 0.1970$ | $92.0840 \pm 0.1965$ | $93.5600 \pm 0.1378$ | 52.29 |
| SIGNSGD | $93.1840 \pm 0.0513$ | $93.2100 \pm 0.2021$ | $93.2320 \pm 0.1979$ | 72.14 |
| SIGNUM | $93.5420 \pm 0.2513$ | $93.7680 \pm 0.0823$ | $93.8160 \pm 0.0940$ | 54.62 |
| ADAMW | $93.6020 \pm 0.2622$ | $93.9880 \pm 0.1066$ | $93.7580 \pm 0.1612$ | 22.02 |
| MUON | $94.4980 \pm 0.1722$ | $94.6740 \pm 0.2110$ | $94.6420 \pm 0.0750$ | 68.16 |

# E. Computational Cost of Non-Euclidean GNS Estimation

This section reports the cost of the $\ell_1$ and $\mathcal{S}_1$ GNS estimators. We first present wall-clock measurements for Llama 3 at the 160M and 1B scales, then give a complexity analysis of the additional compute, communication, and memory costs.

## E.1. Wall-Clock Overhead

We measured the latency of each training step for the 160M and 1B Llama 3 models on a single 8×H100 (80GB) node using the same DDP configuration as our main experiments. Table 12 reports averages over 1,000 optimizer steps with the GNS estimator invoked every $F = 100$ steps, as in Table 5. The cost of the additional reductions, the sample variance computation, the matrix square root, and the nuclear-norm evaluation is amortized over $F$ training steps.

*Table 12.* Average latency of each training step with and without GNS estimation on a single 8×H100 (80GB) node with DDP enabled, averaged over 1,000 optimizer steps with $F = 100$.

| Configuration | Baseline (s/step) | + $\ell_1$ GNS (s/step) | Overhead | + $\mathcal{S}_1$ GNS (s/step) | Overhead |
|---|---|---|---|---|---|
| 160M (local BS 16) | 0.447 | 0.448 | +0.2% | 0.509 | +13.9% |
| 1B (local BS 8) | 0.622 | 0.623 | +0.3% | 0.869 | +39.9% |

Computing the $\ell_1$ GNS adds an additional element-wise square operation with one additional `AllReduce` over the per-element statistics. This adds under $1\%$ overhead at both scales. However, computing the $\mathcal{S}_1$ GNS incurs a larger overhead, particularly at the 1B scale, since it requires computing Gram matrices and an eigendecomposition for each parameter tensor. We leave further memory and performance optimizations of our implementation (FSDP-aware sharding of the Gram matrices, employing efficient iterative matrix-square-root algorithms, e.g., coupled Newton, and GNS computation and communication overlap) to future work.

## E.2. Theoretical Cost Breakdown

We summarize the computational, communication, and memory complexities associated with our $\ell_1$ and $\mathcal{S}_1$ GNS estimators in Table 13. This is based on pseudocode compatible with FSDP provided in Algorithm 2 and 3 for the $\ell_1$ and $\mathcal{S}_1$ GNS estimators, respectively. For our analysis, let $P_l = m_l n_l$ (assuming $m_l \leq n_l$) denote the parameter count for layer $l$, $P = \sum_l P_l$ denote the total parameter count, and $R$ denote the number of data-parallel ranks.

**Notation for Distributed Collectives.** Throughout the algorithms, following Ahn et al. (2025), distributed collectives are written as explicit tensor shape transformations:

- `AllGather`: restores a full tensor from parameter shards.

- `ReduceScatter`: reduces across data-parallel ranks and returns a sharded tensor.

- `AllReduce`: reduces across data-parallel ranks and broadcasts the result to all ranks.

*Table 13.* Summary of additional computational, communication, and peak memory overheads for the proposed non-Euclidean GNS estimators ($\ell_1$ and $\mathcal{S}_1$) under standard Distributed Data Parallel (DDP) and Fully Sharded Data Parallel (FSDP) configurations. Costs are reported per measurement step and all memory buffers are transient.

| Overhead Type | Distribution | $\ell_1$ GNS Estimator | $\mathcal{S}_1$ GNS Estimator |
|---|---|---|---|
| **Compute** (Per Layer) | DDP / FSDP | $\mathcal{O}(P_l)$ element-wise squares and variance computation | $\mathcal{O}(m_l^2 n_l)$ Gram matrix matmul $+ \mathcal{O}(m_l^3)$ eigendecomposition $+ \mathcal{O}(m_l^2 n_l)$ mean-Gram computation |
| **Communication** (Per Measurement) | DDP | $\mathcal{O}(P)$ `AllReduce` for squared-gradient statistics | $\mathcal{O}(\sum_l m_l^2)$ `AllReduce` for Gram matrix aggregation |
| | FSDP | $\mathcal{O}(P)$ `ReduceScatter` for squared-gradient statistics $+ \mathcal{O}(1)$ scalar `AllReduce` | $\mathcal{O}(\sum_l m_l^2)$ `AllReduce` for Gram matrix Aggregation $+ \mathcal{O}(P)$ `AllGather` for full mean gradient reconstruction |
| **Peak Memory** (Per Layer) | DDP | $\mathcal{O}(P_l)$ squared-gradient buffer | $\mathcal{O}(m_l^2)$ Gram matrix buffer |
| | FSDP | $\mathcal{O}(P_l/R)$ sharded squared mean-gradient buffer $+ \mathcal{O}(P_l)$ squared-gradient buffer | $\mathcal{O}(m_l^2)$ Gram matrix buffer $+ \mathcal{O}(P_l)$ reconstructed full mean-gradient buffer |

---

**Algorithm 2** $\ell_1$ GNS Estimation with FSDP at step $k$

---

**Require:** $R$ data-parallel ranks; global batch size $B_k$; $L$ layers with parameters $x_{k,l} \in \mathbb{R}^{m_l \times n_l}$, $P_l = m_l n_l$;

1: **for** $l = L, L-1, \ldots, 1$ **(reverse layer order, backward pass) do**
2:     **// — FSDP Backward —**
3:     `AllGather`$(x_{k,l}^j) \rightarrow x_{k,l}$        {Restore full parameters on rank $j$}
4:     Compute local unsharded gradient: $g_{k,l}^j = \dfrac{R}{B_k} \sum_{\xi \in \mathcal{S}_k^j} \nabla_{x_{k,l}} \ell(x_k; \xi)$
5:     **// — Pre-ReduceScatter Hook: Per-Layer Variance Statistics —**
6:     Compute local element-wise square: $z_{k,l}^j \leftarrow (g_{k,l}^j)^2$        {$O(P_l)$ element-wise ops; rank-local only}
7:     **// — FSDP Gradient Sharding —**
8:     $\bar{g}_{k,l} \leftarrow$ `ReduceScatter`$\left(\frac{1}{R} \sum_{j=1}^R g_{k,l}^j\right)$        {Standard FSDP; shard size $P_l/R$}
9:     $h_{k,l} \leftarrow$ `ReduceScatter`$\left(\frac{1}{R} \sum_{j=1}^R z_{k,l}^j\right)$        {1 additional RS; shard size $P_l/R$}
10:    **// — Post-ReduceScatter: Compute Variance —**
11:    Compute per-coordinate variance for shard $j$ (Equation (20)):

$$(\hat{\sigma}_{k,l}^j)^2 = \frac{B_k}{R-1}\left(h_{k,l}^j - (\bar{g}_{k,l}^j)^2\right),$$

12:     Store $\ell_1$ norm for the variance of layer $l$ shard $j$: $\|\hat{\sigma}_{k,l}^j\|_1$
13: **end for**
14: **// — Global Aggregation —**
15: Accumulate layer contributions: $\|\hat{\sigma}_k^j\|_1 = \sum_{l=1}^L \|\hat{\sigma}_{k,l}^j\|_1$, $\|\bar{g}_k^j\|_1 = \sum_{l=1}^L \|\bar{g}_{k,l}^j\|_1$.
16: $\|\hat{\sigma}_k\|_1 \leftarrow$ `AllReduce`$\left(\sum_{j=1}^R \|\hat{\sigma}_k^j\|_1\right)$        {All reduce a scalar across shards}
17: $\|g_k\|_1 \leftarrow$ `AllReduce`$\left(\sum_{j=1}^R \|\bar{g}_k^j\|_1\right)$        {All reduce a scalar across shards}
18: Compute $\ell_1$ GNS: $\hat{\mathcal{B}}_{\ell_1} = \frac{\|\hat{\sigma}_k\|_1^2}{\|g_k\|_1^2}$.

---

---

**Algorithm 3** $\mathcal{S}_1$ GNS Estimation with FSDP at step $k$

---

**Require:** $R$ data-parallel ranks; global batch size $B_k$; $L$ layers with parameters $X_{k,l} \in \mathbb{R}^{m_l \times n_l}$, $m_l \le n_l$, $P_l = m_l n_l$

1: **for** $l = L, L{-}1, \dots, 1$ **(reverse layer order, backward pass) do**

2:     **// — FSDP Backward —**

3:     $\texttt{AllGather}(X_{k,l}^j) \rightarrow X_{k,l}$               {Restore full parameters on rank $j$}

4:     Compute local unsharded gradient matrix: $G_{k,l}^j = \dfrac{R}{B_k} \sum_{\xi \in \mathcal{S}_k^j} \nabla_{X_{k,l}} \ell(X_k; \xi)$

5:     **// — Pre-ReduceScatter Hook: Per-Layer Gram Statistics —**

6:     Compute local Gram matrix: $Q_{k,l}^j \leftarrow G_{k,l}^j \big(G_{k,l}^j\big)^\top \in \mathbb{R}^{m_l \times m_l}$      {$O(m_l^2 n_l)$ matmul; rank-local only}

7:     Aggregate Gram matrices across ranks: $A_{k,l} \leftarrow \texttt{AllReduce}\big(\sum_{j=1}^R Q_{k,l}^j\big)$      {$m_l^2$ elements; $m_l^2 \ll P_l$}

8:     **// — FSDP Gradient Sharding —**

9:     $\bar{G}_{k,l} \leftarrow \texttt{ReduceScatter}\big(\frac{1}{R}\sum_{j=1}^R G_{k,l}^j\big)$      {Standard FSDP; shard size $P_l/R$}

10:    **// — Post-ReduceScatter: Compute Row-Covariance —**

11:    Reconstruct full mean gradient: $\bar{G}_{k,l} \leftarrow \texttt{AllGather}(\bar{G}_{k,l}^j)$      {$P_l$ elements; GNS-specific AllGather}

12:    Compute mean Gram matrix: $B_{k,l} \leftarrow \bar{G}_{k,l}\bar{G}_{k,l}^\top \in \mathbb{R}^{m_l \times m_l}$

13:    Compute unbiased row-covariance proxy (Equation (21)):

$$C_{\text{row},k,l} = \frac{B_k}{R-1}\left(\frac{1}{R} A_{k,l} - B_{k,l}\right)$$

14:    Compute and store layer contributions: $\|C_{\text{row},k,l}^{1/2}\|_{\mathcal{S}_1}$, $\|\bar{G}_{k,l}\|_{\mathcal{S}_1}$ {$O(m_l^3)$ eigendecompositions; local on each rank}

15: **end for**

16: **// — Global Aggregation —**

17: Accumulate layer contributions: $\|C_{\text{row},k}^{1/2}\|_{\mathcal{S}_1} = \sum_{l=1}^L \|C_{\text{row},k,l}^{1/2}\|_{\mathcal{S}_1}$ , $\|G_k\|_{\mathcal{S}_1} = \sum_{l=1}^L \|\bar{G}_{k,l}\|_{\mathcal{S}_1}$.

18: Compute $\mathcal{S}_1$ GNS: $\hat{\mathcal{B}}_{\mathcal{S}_1} = \frac{\|C_{\text{row},k}^{1/2}\|_{\mathcal{S}_1}^2}{\|G_k\|_{\mathcal{S}_1}^2}$.

---

