# OpenReview forum: "Adaptive Batch Sizes Using Non-Euclidean Gradient Noise Scales for Stochastic Sign and Spectral Descent"
_ICML.cc/2026/Conference — ICML 2026 regular_

### Official Review · Reviewer_Nk47 · 2026-03-10

**Soundness:** 3
**Presentation:** 3
**Significance:** 4
**Originality:** 3
**Overall Recommendation:** 4
**Confidence:** 2

**Summary:**

This paper studies adaptive batch size selection for non-Euclidean optimizers, focusing on sign-based (signSGD/Signum) and spectral-based (specSGD/Muon) methods. The central claim is that the classical Euclidean gradient noise scale (GNS), which is often used to decide critical batch size (CBS), does not match the geometry of these optimizers. Instead, the noise should be measured in the dual norm of the optimizer.

The authors derive new GNS formulas in non-Euclidean geometry: an L1-based GNS for sign methods and a nuclear(S1)-norm GNS for spectral methods. Starting from lower bounds on expected single-step improvement under smoothness assumptions, they show that the optimal batch size is proportional to the ratio of noise to signal measured in the correct dual norm. This leads to a practical rule of the form $B_k \propto \theta^{-2}\mathcal{B}(x_k)$, where $\theta$ is a tunable constant.

To make this usable in large-scale distributed training (DDP/FSDP), the paper proposes a low-overhead estimator. It uses per-rank gradients as independent samples and computes signal and noise statistics with small additional communication. An adaptive algorithm is introduced with smoothing, monotonic batch growth after warmup, and $\sqrt{B_k}$ learning-rate scaling.

Experiments are conducted on language models (Llama-3 160M and 1B on C4) and vision tasks. The method significantly reduces the number of optimizer steps (up to around 60% in some cases) while maintaining similar or slightly better validation loss.

Overall, the paper argues that measuring GNS in the optimizer's natural geometry gives a more principled adaptive batch rule for non-Euclidean methods.

**Compliance With Llm Reviewing Policy:**

Affirmed.

**Key Questions For Authors:**

1. Sensitivity to $\theta$: How sensitive are the results to the choice of $\theta$ across different models and datasets? If the same $\theta$ works well in all settings, this would strengthen the practical value of the method.
2. Overhead analysis: Can the authors provide detailed numbers for additional wall-clock time, communication cost, and memory usage introduced by L1 and S1 GNS estimation, especially in the largest experiments?
3. Can the authors isolate the effect of geometry-aware GNS from $\sqrt{B_k}$ learning-rate scaling and monotonic batch growth? This would clarify how much improvement comes directly from the proposed GNS formulation.

**Limitations:**

The paper discusses some limitations but could expand further. The practical rule depends on tuning $\theta$. In addition, more detailed reporting of communication and memory overhead would improve transparency.

**Strengths And Weaknesses:**

(1a) Soundness - Strengths: The theoretical part is structured clearly. The extension from Euclidean GNS to dual-norm GNS is reasonable and follows standard arguments. The assumptions (smoothness, quadratic approximation) are standard in optimization papers. The experiments are done on realistic large-scale workloads, including 1B parameter language models. The results report both step reduction and validation loss, which is appropriate. The distributed estimator is described clearly enough to understand how it works in practice.
(1b) Soundness - Weaknesses: The theory is based on lower bounds, so the constants are loose. In practice, the rule depends on a tunable parameter $\theta$. The paper does not deeply study how sensitive results are to this parameter. It is unclear how robust the method is without tuning. The estimator is described as low overhead, but there is no detailed quantitative breakdown of extra wall-clock time, communication cost, or memory usage. This is especially important for the nuclear norm case in large models. Overall, the work seems technically correct, but more analysis of robustness and overhead would strengthen it.

(2a) Presentation - Strengths: The paper is generally well organized. The flow from theory to estimator to experiments is easy to follow. The idea of using the dual norm to measure noise is clearly explained. Tables and figures support the claims.
(2b) Presentation - Weaknesses: Some design choices (e.g., value of $\theta$, smoothing factors, monotonic growth rule) are stated but not deeply analyzed. A small sensitivity study in the main text would help.

(3) Significance: Adaptive batch sizing is an important problem in large-scale training. Extending GNS analysis beyond Euclidean SGD is timely and relevant, especially as sign and spectral optimizers are becoming more popular. The reported reduction in optimization steps is significant and can reduce training cost and energy usage. The work is likely to be useful for practitioners working on large distributed training. The theoretical contribution is incremental, but the geometric extension plus the practical estimator make it useful in practice.

(4) Originality: The main originality is the idea of defining gradient noise scale in the dual norm of the optimizer and deriving explicit L1 and S1 forms for sign and spectral methods. This gives a clean geometric interpretation across different optimizers. The estimator and scheduling algorithm are more engineering contributions, but they are well integrated and practically useful. The paper does not introduce a completely new optimization method, but it extends existing ideas in a reasonable and useful way.

---

> ### Author Rebuttal · Authors · 2026-03-30
>
> Thank you for your questions and comments. We address each concern below.
>
> ### __1. Sensitivity to $\theta$__
>
> The following table presents our $\theta$ ablation for signSGD for the Llama 160M model averaged over 10 seeds. We observe that different $\theta$ still achieve better validation loss than:
> 1. the best constant-batch baseline (3.9357 in Table 2), and
> 2. the $\ell_2$-based adaptive batch-size (3.7431 in Table 9).
>
> | $\theta$ | 0.5 | 0.6 | 0.7 |
> |---|---|---|---|
> | Validation loss | 3.6909 ± 0.0097 | **3.6817 ± 0.0094** | 3.7292 ± 0.0092 |
> | Total training steps | 7650 | 9950 | 12374 |
>
> Additionally, all Llama experiments (across all algorithms) in the main paper use a fixed $\theta=0.6$. This value was tuned once on SignSGD, then transferred to other settings, without re-tuning for other algorithms and model scales (160M to 1B). These results strengthen the practical value of the method. If accepted, we will further emphasize this in our final manuscript.
>
> ### __2. Cost and Overhead Analysis__
>
> We measured wall-clock overhead on 8×H100 (DDP), computing GNS every $F{=}100$ steps.
>
> | Config | Baseline (s/step) | + $\ell_1$ GNS | Overhead | + $\mathcal{S}_1$ GNS | Overhead |
> |---|---|---|---|---|---|
> | 160M (local BS 16) | 0.447 | 0.448 | **+0.2%** | 0.509 | **+13.9%** |
> | 1B (local BS 8) | 0.622 | 0.623 | **+0.3%** | 0.869 | **+39.9%** |
>
> The $\ell_1$ GNS is effectively free with <1% overhead. $\mathcal{S}_1$ GNS incurs much higher overhead due to the matrix square root and nuclear norm computation, which we currently compute using an eigendecomposition. However, we believe that this can be further optimized through standard performance optimizations (e.g., CPU offload, overlap with forward pass, FSDP-aware sharding) and better numerical linear algebra methods (coupled Newton, etc.); we leave these to future work as they are orthogonal to the algorithmic contribution of this paper.
>
> Theoretically, overhead of GNS estimation is also negligible in compute, communication, and memory.
> For each layer $\ell$, let $G_\ell \in \mathbb{R}^{m \times n}$, $P_\ell = mn$, $d = \min(m,n)$, $D = \max(m,n)$, and $s$ denote the sequence length.
>
> #### __Compute__
>
> | | Added cost | Ratio to backward $\bigl(\mathcal{O}(P_\ell \cdot s)\bigr)$ |
> |---|---|---|
> | $\ell_1$/$\ell_2$ GNS | $\mathcal{O}(P_\ell)$ — element-wise ops | $\mathcal{O}(1/s)$ |
> | $\mathcal{S}_1$ GNS | $\mathcal{O}(d^2 D)$ + $\mathcal{O}(d^3)$ — Gram + eigendecomposition | $\mathcal{O}(d/s)$ |
>
> #### __Communication__
>
> | | Added volume | Collective | Frequency |
> |---|---|---|---|
> | $\ell_1$/$\ell_2$ GNS | $P_\ell$ | 1 extra ReduceScatter | $F$-th step only |
> | $\mathcal{S}_1$ GNS | $d^2$ (Gram AR) + $P_\ell$ (grad AG) | AllReduce (Gram) + AllGather(grad) | $F$-th step only |
>
> For reference, baseline FSDP already performs AllGather $\times 2$ + ReduceScatter $\times 1$ of size $P_\ell$ every step. Both GNS overheads are strictly smaller and amortized.
>
> #### __Memory__
>
> | | Peak | Persistent |
> |---|---|---|
> | $\ell_1$/$\ell_2$: $g^{\odot 2}$ buffer | $\mathcal{O}(P_\ell)$ — transient, freed immediately after use | $O(1)$ |
> | $\mathcal{S_1}$: Gram $G_\ell {G_\ell}^\top$  | $\mathcal{O}(d^2) + \mathcal{O}(P_\ell)$ ${}^\dagger$ — freed immediately after use | $O(1)$ |
>
> $\dagger$: This $P_\ell$ from AG of $\bar{G}_\ell$
>
> In summary, $\ell_1$ GNS estimation is lightweight across all three dimensions. $\mathcal{S}_1$ GNS requires non-trivial compute and communication per measurement, but still has net training efficiency gains, although there remains further room for practical improvement.
>
> ### __3. Isolating Geometry-Aware GNS from LR Scaling and Monotonic Growth__
>
> Agreed. The effect of learning-rate scaling has been ablated in the Appendix D, Table 8. The results show that learning-rate scaling yields a measurable gain, but only accounts for about 1% of the total gains achieved in validation loss over the constant batch size baseline.
>
> The monotonic batch-size growth condition is primarily a safeguard against oscillations induced by noisy GNS estimates. Given that our objective is to increase batch sizes to maximize hardware efficiency without sacrificing model quality, there is little motivation to decrease the batch size if one achieves good validation loss. Moreover, as shown by the smoothed GNS trajectories in Figure 1 and Appendix Figure 3, the trend is predominantly increasing, which suggests that the monotonic constraint rarely overrides the natural batch size trend. Lastly, we also expect as one increases the number of data-parallel workers, the GNS estimate will become more accurate, which will diminish the need for additional smoothing in practice.
>
> We will add these clarifications and emphasize this ablation in the main text in the final manuscript.

---

> > ### Author Rebuttal · Reviewer_Nk47 · 2026-04-03
> >
> > The rebuttal is helpful, and the additional experimental results (especially sensitivity to $\theta$ and overhead analysis) are appreciated.
> >
> > While some aspects could be explored further, the contribution is clearer after the rebuttal. Overall, I keep my original assessment and score unchanged.

---

> > > ### Author Response · Authors · 2026-04-07
> > >
> > > We sincerely thank the reviewer for their thoughtful engagement and for confirming that all concerns have been fully resolved. We are glad the additional results on $\theta$ sensitivity and overhead analysis were helpful in clarifying the contribution. If there are any remaining aspects where the paper could be further strengthened, we would welcome specific suggestions and are happy to incorporate them into the final manuscript.

---

### Official Review · Reviewer_WgPW · 2026-03-11

**Soundness:** 3
**Presentation:** 2
**Significance:** 3
**Originality:** 3
**Overall Recommendation:** 4
**Confidence:** 4

**Summary:**

The paper proposes a method that can choose adaptive batch sizes during training. This paper points out that existing Gradient Noise Scale (GNS) methods determine optimal batch sizes considering Euclidean geometry, which causes a mismatch when applied to non-Euclidean optimizers like signSGD and Muon. The authors extend GNS to non-Euclidean cases (both $\ell_\infty$ and $\mathcal{S}_\infty$) that fit into Signum and Muon. For computational efficiency, the authors propose a scalable variance estimation framework that leverages local mini-batch gradients across distributed data-parallel ranks. Empirical results on language and vision tasks demonstrate consistent improvement of this batch-size adaptive method over fixed batch sizes.

**Compliance With Llm Reviewing Policy:**

Affirmed.

**Final Justification:**

This paper claims their adaptive method can be used for both **sign descent** and **spectral descent** as shown in the title. I'm convinced about the sign descent part, but still have concerns about the spectral descent part due to 1) the lack of a larger-scale experiment (only includes 160M models) and 2) the computational overhead cannot be ignored. Thus, I think this paper is still on the borderline.

I recommend:
1) including one more experiment on a larger scale. If the authors can't afford 1B, maybe ~500M run also helps.
2) In Table 2-4, change "Steps reduction" to "Wall-clock time reduction" (or add a column). The current comparison is not straightforward regarding actual improvement.

**Key Questions For Authors:**

1. Could you please include SpecSGD and Muon for Llama 1B model in Table 3?
2. Do you have hypotheses regarding the worse performance than AdamW baseline in Table 3 and 10?
3. Can you provide wall-clock time and computational cost analysis?
4. How sensitivity of this method to the hyperparameter $\theta$?
5. Can you apply this to Scion [1]? Scion uses Signum for embedding layer and Spectral descent for hidden layers, which should be a perfect test bed for this work considering both $\ell_\infty$ and $\mathcal{S}_\infty$.


[1] Pethick, T., Xie, W., Antonakopoulos, K., Zhu, Z., Silveti-Falls, A., and Cevher, V. Training deep learning models with norm-constrained lmos. ICML, 2025.

**Limitations:**

It should include a discussion about computational overhead.

**Strengths And Weaknesses:**

**Strengths:**
1. The motivation is interesting. The authors notice that applying Euclidean-based GNS metrics to non-Euclidean optimizers is not feasible.
2. The paper derives the GNS metrics based on the dual norms ($l_1$ for sign-based, $\mathcal{S}_1$ for spectral), which provides a highly principled foundation for the proposed algorithms.
3. The paper considers efficiency in practical implementation as well. It utilizes the local mini-batch gradients already computed across independent ranks in DDP and FSDP.
4. The experiments cover both language and image tasks and show promising results.

**Weaknesses:**
1. Could you please include SpecSGD and Muon for Llama 1B model in Table 3? The results of 160M models shown in Table 2 looks promising. But validation on larger models like 1B is also necessary for spectral descent.
2. In Table 3, the adaptive AdamW strategy fails to match the baseline validation loss when scaling up to the 1B model. Same observation in Table 10. Do you have hypotheses regarding it fails?
3. Miss wall-clock time and computational cost analysis. Especially for spectral descent, the proposed method requires reconstructing the full covariance matrix from local gram matrices. The discussion about the additional cost is necessary.
4. How sensitivity of this method corresponding to the hyperparameter $\theta$?

---

> ### Author Rebuttal · Authors · 2026-03-30
>
> Thanks for your comments! We address each concern below.
>
> ### __1. SpecSGD/Muon for Llama-1B (Table 3)__
>
> While we recognize the value of 1B-scale validation for spectral methods, conducting these experiments with sufficient statistical power (e.g., multiple seeds and hyperparameter tuning) is not possible due to the experiments exceeding our current time and computational budget. If we obtain additional compute resources, we may add the results in the final manuscript.
>
> ### __2. Adaptive AdamW Fails at 1B Scale__
>
> We appreciate the observation and hypothesize that the failure of adaptive AdamW at 1B scale is related to the derived $\ell_1$ GNS for sign-based methods not capturing the effective geometry of Adam. While Adam can be viewed as a sign-based method, it deviates significantly in practice through its use of smoothed first- and second-moment estimates. As a result, Adam does not fully align with the signSGD geometry (A Orvieto and R. M. Gower, NeurIPS 2025), and there remains a gap between the optimizer and the geometry assumed by the GNS.
>
> We view this mismatch as an important limitation; bridging this gap is a vital direction for future work. We will discuss this explicitly in the final manuscript.
>
> ### __3. Wall-Clock Time and Computational Cost__
>
> Please refer to our response to Reviewer Nk47 below.
>
> ### __4. Sensitivity to $\theta$__
>
> Please refer to our response to Reviewer Nk47 below.
>
> ### __5. Application to Scion__
>
> We appreciate the suggestion and agree that Scion represents a compelling test bed for our framework. While our current GNS metrics provide a rigorous treatment for Signum and SpecSGD independently, extending these non-Euclidean geometries to a composite optimizer setting remains an open theoretical and practical question. We agree that investigating these interactions in Scion is the natural next step for generalizing composite GNS metrics, and we will highlight this direction in our conclusion.

---

> > ### Author Rebuttal · Reviewer_WgPW · 2026-04-04
> >
> > Thanks for the authors’ reply, especially for the sensitivity of $\theta$ and computational overhead.
> >
> > I understand the restriction of computational resources. However, I still think the experiment of language task for spectral method only relying on 160M models is insufficient.
> >
> > In addition, compared with computational overhead of spectral method (13.9% on 160M models), the step reduction 16.49% for SpecSGD reported in Table 2 is marginal.  Moreover, the overhead for 1B models is 39.9% which can’t be ignored. This further indicates the necessity of additional 1B experiments for spectral methods.

---

> > > ### Author Response · Authors · 2026-04-07
> > >
> > > Thanks for your questions! We address each of your comments below.
> > >
> > > > In addition, compared with computational overhead of spectral method (13.9% on 160M models), the step reduction 16.49% for SpecSGD reported in Table 2 is marginal. Moreover, the overhead for 1B models is 39.9% which can’t be ignored. This further indicates the necessity of additional 1B experiments for spectral methods.
> > >
> > > While we agree that the computational overhead for spectral methods is a concern, the proposed method also achieves an improved validation loss over the constant batch baseline in addition to the step reduction of 16.46% reported for specSGD in Table 2 for the 160M model (shown below). For Muon, the step reduction of 66.77% compensates for the required GNS computation overhead.
> > >
> > > | Optimizer | Constant batch | Non-Euclidean GNS |
> > > |---|---:|---:|
> > > | specSGD | 3.3748 ± 0.0096 | **3.3728 ± 0.0094** |
> > >
> > > Moreover, we believe that the spectral GNS computation can be further optimized through standard performance optimizations (e.g., overlap with forward pass, FSDP-aware sharding) and better numerical linear algebra methods (coupled Newton, etc.), which we leave to future work.
> > >
> > > > I understand the restriction of computational resources. However, I still think the experiment of language task for spectral method only relying on 160M models is insufficient.
> > >
> > > While we agree expanding our spectral descent experiments to larger scale is beneficial, the associated compute costs exceed our current constraints. We believe that the combination of these results (including the results above) provide sufficient validation of the proposed approach for spectral methods.

---

### Official Review · Reviewer_se1v · 2026-03-13

**Soundness:** 3
**Presentation:** 3
**Significance:** 3
**Originality:** 3
**Overall Recommendation:** 5
**Confidence:** 4

**Summary:**

In this paper, the authors study adaptive batch-size selection for signSGD/Signum and Spectral Descent/Muon based on the geometry of their corresponding norms. They provide theoretical derivations for the batch-size selection and establish convergence guarantees for the associated step-size and batch-size choices. Finally, they support their theoretical findings with empirical evaluations.

**Compliance With Llm Reviewing Policy:**

Affirmed.

**Final Justification:**

In my initial review, I found the paper's core idea—deriving batch-size and step-size selections based on the geometry of norms—to be mathematically interesting and well-motivated. However, I raised three critical concerns: a discrepancy in the parameter selection for the non-convex analysis, a dimensional inconsistency in the resulting convergence bounds, and a missing critical baseline (Normalized SGD).

The authors provided a rebuttal that directly resolved all of my concerns. By substituting the optimal GNS-derived step size (scaled by the true gradient norm), they derived a tighter bound that aligns the non-convex analysis with the main text. Importantly, this gradient-scaled formulation naturally fixed the dimensional inconsistency I pointed out, removing the awkward theoretical artifact present in their previous constant-step-size bound. Finally, they explicitly acknowledged the mathematical equivalence of their $\ell_2$ method to NSGD and successfully ran the missing NSGD baselines for their vision experiments.

Because the authors have provided the corrected mathematics and the missing empirical baselines during the rebuttal phase, the soundness and clarity of the paper are now significantly improved. Assuming these changes are integrated into the final manuscript, I am happy to recommend this paper for acceptance.

**Key Questions For Authors:**

No Questions

**Limitations:**

No Limitations

**Strengths And Weaknesses:**

### **Strengths**

1. **Presentation & Motivation:** The paper is well-written and well-motivated, clearly articulating the problem it aims to address.
2. **Theoretical Novelty:** The theoretical approach of deriving batch-size and step-size selections based on the geometry of norms is an interesting idea.

### **Weaknesses**

**1. Discrepancy in Non-Convex Analysis:** In the appendix, Theorem A.4 and Theorem A.5 state convergence guarantees for the studied methods using a step-size selection that distinctly differs from the one proposed in the main text. Currently, the theory supporting the main text's parameter selection appears to hold only in the strongly convex case. Could the authors provide convergence guarantees for the explicitly claimed parameter selections in the non-convex setting, or clarify this discrepancy?

**2. Dimensional Inconsistency of the Step-Size and Bounds:** The proposed step-size $\eta = \frac{1}{\sqrt{L_{\infty}K}}$ lacks physical meaning and leads to dimensional inconsistencies in the analysis. Specifically, in Theorems A.4.1 and A.5.1, the bounds for convergence to a stationary point contain the multiplier $(\mathcal{L}(x_0) - \mathcal{L}^{\star} +\frac{1}{2})$. In practical optimization, the objective function $\mathcal{L}$ often carries physical measurement units. Adding a dimensionless constant (like $\frac{1}{2}$) directly to a dimensional quantity breaks standard dimensional analysis. This awkward theoretical artifact appears to occur because the step-size selection logic does not naturally scale with the objective function.

**3. Missing Baseline (NSGD) in Comparisons:** Table 1 provides a comparison between SGD, signSGD, and specSGD. To make this comparison complete and rigorously contextualized, the authors must include Normalized SGD (NSGD). This addition is necessary because the method presented in Equation (2), when evaluated under $\ell_2$-geometry, is mathematically equivalent to NSGD.

---

> ### Author Rebuttal · Authors · 2026-03-30
>
> Thanks for your comments! We address each concern below.
>
> ### __Discrepancy in Non-Convex Analysis__
>
> Thank you for highlighting this discrepancy. We agree that it is crucial to analyze the non-convex guarantees under the parameterization stated in the main text. Under the *optimal* GNS-derived step size, one obtains a sharper result because the step size is scaled by the true gradient norm. For signSGD, combining the step size in (12) with the batch-size prescription in Theorem A.4 yields the step size $
> \eta_k = (1-\theta) \frac{\Vert \nabla \mathcal{L}(x_k) \Vert_1}{L_\infty}$ (replacing $d$ scaling with $L_\infty$ for simplicity). Note that $\theta$ appears here since it bounds the square root of the $\ell_1$ GNS, which is the quantity that appears in the main text.
>
> Substituting this choice into the descent inequality (25) yields:
> $$\frac{1}{K}\mathbb{E}\left[\sum_{k=0}^{K-1} \Vert \nabla\mathcal{L}(x_k) \Vert_1^2\right] \le \frac{2L_\infty}{(1-\theta)^2K} \big(\mathcal{L}(x_0)-\mathcal{L}^*\big).$$
> This bound is stronger than Theorem A.4.1 for convergence to a stationary point, and an analogous guarantee can be derived for specSGD. We will incorporate these results in Theorems A.4 and A.5 in the final manuscript.
>
> ### __Dimensional Inconsistency of Step-Size__
>
> Thank you for highlighting this inconsistency. We agree that the extra additive constant in Theorems A.4.1/A.5.1 is an artifact of the specific fixed-step-size choice $\eta=\frac{1}{\sqrt{KL}}$.
>
> Our intent in including this case is to align with the standard constant-batch non-convex analysis established in prior signSGD work (e.g., Bernstein et al., 2018, Theorem 1), where a similar term also appears. Incorporating the batch size rule with the constant step size $\eta=\frac{1}{\sqrt{KL}}$ dissolves the variance dependent term $\|\sigma_k\|_1$ from the neighbourhood size, yielding a tighter convergence bound.
>
> This dimensionality issue disappears when the step size is scaled with the gradient-norm scaling (as in our GNS-derived rule $\eta_k = (1-\theta) \frac{\Vert \nabla \mathcal{L}(x_k) \Vert_1}{L_\infty}$), as noted in the above response. This is consistent with the interpretation of sign-based methods as unnormalized steepest descent under the $\ell_\infty$ geometry (see Balles, Pedregosa, and Le Roux, 2020, Table 1). We will clarify this point in the final manuscript and emphasize the gradient-scaled step-size result.
>
> ### __Missing NSGD Baseline__
> We appreciate the observation. The $\ell_2$-geometry row in Table 1 yields $p_k = g_k/‖g_k‖_2$, which is indeed normalized SGD (NSGD). We will make this explicit in the table caption and discussion.
>
> Regarding empirical comparison, we will add NSGD as a baseline in the Imagewoof experiments. Since SGD-based methods are generally not used for training large language models such as Llama, we do not include SGD/NSGD experiments in the language modeling setting. Instead, we include this comparison in the Imagewoof workload, where SGD-based baselines are standard and the evaluation is more appropriate.
>
> | Optimizer | ${\mathcal{S}_1}$ GNS | $\ell_2$ GNS | $\ell_1$ GNS |
> | :--- | :--- | :--- | :--- |
> | **Normalized SGD** | 1.9814 ±0.000024 | **1.2347 ±  0.016564** | 1.9758 ±0.000198 |

---

> > ### Author Rebuttal · Reviewer_se1v · 2026-04-04
> >
> > Thank you for the detailed response. I appreciate the core concept of this paper. Assuming the authors add the new proofs and discussions from the rebuttal phase to the main text, I am happy to support this work. I have raised my score to a 5 (Accept).

---

> > > ### Author Response · Authors · 2026-04-07
> > >
> > > Thank you again for your in-depth review of our work!

---

### Official Review · Reviewer_NFzz · 2026-03-16

**Soundness:** 2
**Presentation:** 3
**Significance:** 2
**Originality:** 2
**Overall Recommendation:** 4
**Confidence:** 4

**Summary:**

The paper introduces gradient noise scale that accounts for the geometry induced by the optimizer, deriving expressions for non-euclidian GNS and CBS. The authors propose a practical way to estimate those quantities in a distributed regime. They then conduct a number of experiments confirming the advantage of the batch size scheduling based on those quantities compared to constant batch sizes.

**Compliance With Llm Reviewing Policy:**

Affirmed.

**Final Justification:**

The rebuttal addressed my initial concerns, so I am raising my overall score from 3 to 4. That said, the limited large-scale validation - along with the possible lack of robustness at that scale, as suggested by the AdamW results - still makes the paper borderline.

**Key Questions For Authors:**

- Could you please specify the wall-clock computational cost to the CBS to understand its practical feasibility
- Although there is no proper theory of how to incorporate e.g. momentum, there are still speed-ups for Muon. Do you think there needs to be a significant modification to the GNS definition in the presence of moving averages (including for e.g. Shampoo)?

**Limitations:**

yes

**Strengths And Weaknesses:**

Strengths

- Consistent derivations of non-euclidian GNS, with a clever lower-bounding trick. Overall, the geometric argument is clean and intuitive. Some of the experimental results (e.g. for Muon vs constant batch size) are quite impressive.
- The paper covers both text and image datasets
- The paper provides an efficient implementation for the non-euclidian CBS, accounting for distributed setting

Weaknesses

- Essentially, I will only list one weakness, but I think it is unfortunately strong enough to warrant rejection. The main weakness is a lack of comparison against an adaptive batch size scheduling governed by euclidian norm ($\mathcal B_{\ell_2}$) in the paper’s notation. In particular, in the paper, this comparison is only done in the case of SignSGD, which is the worst-performing optimizer in your testing by a significant margin, and only with the smallest of your settings, 160M Llama. For all other optimizers (including, Muon, being the most practically interesting) and settings, there is only comparison against the constant batch size, where the improvement is to be expected. Lacking a comparison with the euclidian adaptive batch size significantly weakens the claims made in the paper about the usefulness of non-euclidian CBS. Importantly, the theory derivation, although well done, are not sufficient in the case to justify non-euclidian CBS, as there are quite a lot of assumption happening to derive any of the CBSs of the paper - starting from the Hessian being the identity; moreover that the derivation of CBS are fundamentally more of a heuristic rather than theoretically rigorous justifications, thus requiring empirical justification of its efficacy
- The 1B Llama experiments do not report variability over random seeds

---

> ### Author Rebuttal · Authors · 2026-03-30
>
> Thanks for your comments! We address each concern below.
>
> ### __Missing Comparison: Non-Euclidean GNS vs. Euclidean GNS for Muon/Signum__
>
> We agree that comparing against Euclidean GNS is a critical baseline. We have added direct comparisons of Euclidean ($\ell_2$) and non-Euclidean GNS based adaptive batch size across optimizers on the 160M Llama workload (10 seed average):
>
> | Optimizer | Constant batch | Euclidean GNS | Non-Euclidean GNS |
> |---|---:|---:|---:|
> | Signum | 3.3737 ± 0.0045 | 3.3842 ± 0.0055 | **3.3707 ± 0.0026** |
> | MUON | 3.3041 ± 0.0021  | 3.3181 ± 0.0072 | **3.3061 ± 0.0027** |
>
> The Euclidean GNS variant fails to match the constant-BS baseline, while geometry-matched variants consistently improve upon it. The same trend holds on Imagewoof:
>
> | Optimizer | ${\mathcal{S}_1}$ GNS | $\ell_2$ GNS | $\ell_1$ GNS |
> |---|---:|---:|---:|
> | SignSGD | 1.179 ± 0.001 | 1.162 ± 0.001 | **1.129 ± 0.019** |
> | MSGD | 1.219 ± 0.000 | **1.185 ± 0.004** | 1.272 ± 0.000 |
> | Muon | **0.624 ± 0.008** | 0.628 ± 0.000 | 0.628 ± 0.000 |
>
> These results isolate the benefit of geometry-aware GNS and will be added to the final manuscript (accompanied by the results for specSGD and AdamW as they require more time).
>
> ### __Theoretical Assumptions ($H=I$) and Heuristic Nature of CBS__
>
> We acknowledge that the one-step improvement analysis in Section 3 is a heuristic motivation. However, we wish to clarify that our convergence proofs in Appendix A (Theorems A.4 and A.5) do not rely on this assumption. We introduce the $H = I$ assumption in order to simplify our presentation and follows previous work by McCandlish, et al., (2018).
>
> Note that our non-Euclidean derivations require strictly weaker assumptions than the Euclidean GNS; specifically, for L1 GNS, only $\mathbb{E}[\mathrm{sign}(g)^\top H\, \mathrm{sign}(g)] \approx \mathrm{Tr}(H)$ (which cancels in the GNS) is necessary; for nuclear GNS, the Ky Fan bound provides a deterministic bound with no Hessian assumption at all.
>
> ### __1B Llama: No Variability Over Random Seeds__
>
> Thank you for the comment. Below is the 10 seed average result for 1B Llama3 model for signSGD which shows consistent improvement across seeds. Results for Signum and AdamW will also be included in the final manuscript.
>
> | Optimizer | Constant batch | Non-Euclidean GNS |
> |---|---:|---:|
> | signSGD | 3.1417± 0.0047 | **2.9946 ± 0.0201** |
>
> ### __Wall-Clock, Cost, and Overhead Analysis__
>
> Please refer to our response to Reviewer Nk47 below.
>
> ### __GNS with Momentum / Moving Averages (Signum, Muon, Shampoo)__
>
> This is an excellent question. Our current formulation does not account for either the first moment EMA (Signum or Muon) or the preconditioner EMA (AdamW or Shampoo), although we still observe substantial speedups when momentum is present (e.g., for Signum and Muon). Although it is unclear how to do so, we do believe that modifications are necessary to rigorously extend GNS. This is highlighted in our conclusion.

---

> > ### Author Rebuttal · Reviewer_NFzz · 2026-04-04
> >
> > Thank you for the detailed response: the theory explanation and a comparison with Euclidian GNS - this helps support the results.
> >
> > I have some follow-up questions:
> > - I am a bit surprised to see that Euclidian GNS couldn't match constant-size baseline; was it scaling the batch size up more aggressively? I still think the paper would benefit from including batch-size and learning-rate trajectories for these runs, since that would make it easier to interpret why Euclidean GNS underperforms.
> > - My remaining concern is the 1B validation. At that scale, the positive evidence is concentrated on sign-based methods, which do not provide a good baseline: they are unusually sensitive to gradient noise (with smaller batches not always performing better), so they are in some sense the most favorable setting for a geometry-aware GNS. With Muon/specSGD not reported and AdamW not matching the constant-batch baseline, I think stronger large-scale validation is still needed to establish broader practical usefulness.
> >
> >
> > That said, the rebuttal substantially addressed my concerns, so I am raising my overall score to 4.

---

> > > ### Author Response · Authors · 2026-04-07
> > >
> > > Thank you for your response! We address each question below.
> > >
> > > > I am a bit surprised to see that Euclidian GNS couldn't match constant-size baseline; was it scaling the batch size up more aggressively? I still think the paper would benefit from including batch-size and learning-rate trajectories for these runs, since that would make it easier to interpret why Euclidean GNS underperforms.
> > >
> > >
> > > Thank you for your question. The new table below (which includes specSGD) shows that Euclidean GNS is unable to match the constant batch baseline for both Signum and Muon. (Note that this is not the case for SignSGD and SpecSGD.) This suggests that the first-moment EMA introduces noise characteristics which the standard Euclidean GNS fails to capture accurately.
> > >
> > > The corresponding difference in GNS and batch size trends can be seen in Figure 1 for signSGD and [here](https://ibb.co/WvT1H8gd) for Signum. For Signum, we observe that the Euclidean GNS strategy increases the batch size more aggressively compared to the non-Euclidean GNS strategy, driven by a lower $\theta = 0.3$ (which we obtained by tuning $\theta$). We will add this discussion into the paper.
> > >
> > > | Optimizer | Constant batch | Euclidean GNS | Non-Euclidean GNS |
> > > |---|---:|---:|---:|
> > > | Signum | 3.3737 ± 0.0045 | 3.3842 ± 0.0055 | **3.3707 ± 0.0026** |
> > > | specSGD | 3.3748 ± 0.0096 | 3.3743 ± 0.0274 | **3.3728 ± 0.0094** |
> > > | Muon | 3.3041 ± 0.0021  | 3.3181 ± 0.0072 | **3.3061 ± 0.0027** |
> > >
> > > > My remaining concern is the 1B validation. At that scale, the positive evidence is concentrated on sign-based methods, which do not provide a good baseline: they are unusually sensitive to gradient noise (with smaller batches not always performing better), so they are in some sense the most favorable setting for a geometry-aware GNS. With Muon/specSGD not reported and AdamW not matching the constant-batch baseline, I think stronger large-scale validation is still needed to establish broader practical usefulness.
> > >
> > > We will add the following Signum and Adam results for the 1B scale.
> > >
> > > |Optimizer | Constant batch | Non-Euclidean GNS|
> > > |---|---:|---:|
> > > |signSGD|3.1417± 0.0047|2.9946 ± 0.0201|
> > > |Signum|2.8306± 0.0051|2.8354 ± 0.0073|
> > > |AdamW|2.7701± 0.0037|2.8023 ± 0.0049|
> > >
> > > While we agree expanding our spectral descent experiments to larger scale would benefit the paper, the associated costs exceed our current compute and time constraints.
> > >
> > > Regarding AdamW's results, we believe that further work is necessary to extend our GNS metrics to the variance-adapted setting. This limitation will be addressed more clearly in the paper.

---

### Decision · Program_Chairs · 2026-04-30

**Decision:**

Accept (regular)

**Comment:**

This paper is concerned with the adaptive batch size selection for spectral descent methods. Particularly, this work shows that using Euclidean gradient-noise scale (a coefficient that depends on the gradient covariance matrix and gradient norm) to select the batch size does not give satisfactory practical performance and utilizing a non-Euclidean version improves the empirical performance. The authors provide a principled derivation of the new rule which is not implementable and then discussed modifications to make it applicable in practice. Then, the results show improvements in LLM training baselines.

The reviewing team appreciated the principled derivation and the promising empirical results. Reviewer WgPW voiced some valid concerns about the computational overhead of the new mechanism and the lack of experiments with larger models. The authors are recommended to revise according to these comments. Particularly, it will be good to discuss the computational overhead to position their contribution fairly.

The recommendation is acceptance.